# The Role of Vitamin D in Parkinson’s Disease: Evidence from Serum Concentrations, Supplementation, and *VDR* Gene Polymorphisms

**DOI:** 10.3390/neurosci6040130

**Published:** 2025-12-16

**Authors:** Jamir Pitton Rissardo, Ana Leticia Fornari Caprara

**Affiliations:** Neurology Department, Cooper University Hospital, Camden, NJ 08103, USA; fornari-caprara-ana@cooperhealth.edu

**Keywords:** Parkinson’s disease, vitamin D, vitamin D receptor, gene polymorphism, susceptibility

## Abstract

Background/aim: Vitamin D (VitD) has been implicated in neuroprotection, yet its role in Parkinson’s disease (PD) remains unclear. This systematic review and meta-analysis aimed to evaluate the association between VitD status, supplementation, and vitamin D receptor (*VDR*) gene polymorphisms with PD risk and outcomes. Methodology: Following PRISMA guidelines, we searched PubMed, Scopus, and Google Scholar through August 2025 for observational studies, clinical trials, and genetic association studies. Primary outcomes included serum VitD levels in PD versus healthy controls (HCs), prevalence of VitD insufficiency/deficiency, and effects of VitD supplementation on motor symptoms. Secondary outcomes assessed associations between *VDR* polymorphisms and PD susceptibility. Data were synthesized using random- and fixed-effects models, with heterogeneity and publication bias evaluated. PROSPERO (CRD420251133875). Results: Sixty-three studies (*n* ≈ 10,700 participants) met inclusion criteria. PD patients exhibited significantly lower VitD levels (SMD = −0.46; 95% CI: −0.51 to −0.41) and higher odds of insufficiency (OR = 1.52) and deficiency (OR = 2.20) compared to HC. Cohort data suggested sufficient VitD may reduce PD risk (HR = 0.83). Supplementation yielded modest, non-significant improvements in motor outcomes. Among 20 genetic studies, FokI (rs2228570) was most consistently associated with PD, while other *VDR* SNPs showed variable or null associations. Conclusions: VitD deficiency is common in PD and may influence disease risk and motor function. Current evidence indicates limited benefit of supplementation for motor outcomes, and genetic associations remain inconsistent.

## 1. Introduction

PD is a progressive neurodegenerative disorder primarily characterized by the loss of dopaminergic neurons in the substantia nigra pars compacta, leading to striatal dopamine depletion and disruption of basal ganglia circuitry. It is the second most prevalent neurodegenerative disorder worldwide, with its incidence steadily increasing due to aging populations and extended life expectancy [1]. The neurochemical imbalance manifests clinically as cardinal motor symptoms, including bradykinesia, resting tremor, rigidity, and postural instability. Beyond motor dysfunction, PD encompasses a wide spectrum of non-motor features such as cognitive impairment, mood disorders, autonomic dysfunction, and sleep disturbances, which significantly impact quality of life [2].

The etiology of PD is multifactorial, encompassing both genetic predispositions and environmental influences. Recent research has emphasized the importance of modifiable lifestyle and environmental factors in shaping disease risk and progression [3]. Among these factors, nutritional status and particularly vitamin intake has attracted increasing interest from the scientific community and general public. Vitamins are essential bioactive compounds with antioxidant properties that support normal physiological and neurological function. Of particular relevance is VitD, which has emerged as a potential neuroprotective agent (Figure 1) [4].

One of the earliest reports suggesting a link between VitD deficiency and PD in humans was published by Derex et al. in 1997, who at the time considered the association to be rare [5]. In this context, emerging evidence suggests that VitD may play a neuroprotective role by mitigating dopaminergic neuron loss and enhancing both motor and cognitive outcomes [6]. In vitro studies revealed that VitD is related to enhanced mesencephalic cell viability [7] and survival [8] in 6-OHDA PD model. In vivo, its deficiency has been linked to increased risk of PD [9], and one of the hypotheses is related to the VitD’s role in immune modulation [10].

VitD exerts neuroprotective effects through multiple molecular pathways relevant to PD. It modulates calcium homeostasis, reduces oxidative stress by upregulating antioxidant enzymes, and attenuates neuroinflammation via inhibition of NF-κB signaling and suppression of pro-inflammatory cytokines such as TNF-α and IL-6 [11]. Additionally, vitamin D influences apoptotic pathways by promoting anti-apoptotic proteins and reducing caspase activation, thereby supporting neuronal survival. These mechanisms collectively help preserve dopaminergic neurons in the substantia nigra pars compacta, the primary site of degeneration in PD [12]. Importantly, vitamin D receptor (VDR) and activating enzymes such as 1α-hydroxylase are highly expressed in dopaminergic neurons of the substantia nigra and glial cells, suggesting a direct role in motor control and neuroimmune regulation [13]. Dysregulation of VDR signaling, whether through deficiency or genetic polymorphisms (e.g., FokI, BsmI), may impair dopamine synthesis and neuronal survival, contributing to both motor symptoms (bradykinesia, rigidity) and non-motor features (cognitive decline, mood disorders) [14].

Although Wang et al. found no causal relationship between genetically predicted VitD levels and PD using bidirectional Mendelian randomization across three large cohorts, their findings do not eliminate the relevance of VitD in PD research [15]. Observational studies consistently report lower serum VitD levels in PD patients, and VDR are present in dopaminergic neurons, suggesting potential neuroprotective roles [16]. Moreover, VitD may influence PD progression through immune modulation, antioxidant activity, and calcium homeostasis—mechanisms not fully captured by genetic analyses. Therefore, further longitudinal and mechanistic studies are warranted to explore VitD’s diagnostic, prognostic, and therapeutic potential in PD beyond genetic predisposition. This study aims to elucidate the role of VitD in PD by examining its involvement across three key domains: serum concentrations, therapeutic supplementation, and genetic variability.

## 2. Methodology

### 2.1. Primary and Secondary Outcomes

The primary outcome was to (1) compare serum VitD levels between patients with PD and HC. Secondary outcomes included (2) comparisons of the prevalence of VitD deficiency and insufficiency between these groups, as well as the evaluation of the efficacy of VitD supplementation versus placebo on motor symptoms in patients with PD. An additional outcome was the analysis of (3) genotype frequencies of *VDR* polymorphisms and their potential influence on the risk of developing PD.

### 2.2. Literature Search Strategy

This meta-analysis followed the PRISMA statement (Appendix A—PRISMA Statement) [17]. Two reviewers (J.P.R. and A.L.F.C.) systematically searched the electronic PubMed/Medline, Scopus, and Google Scholar databases by the following search terms: vitamin D, calciferol, and Parkinson’s disease (Appendix A—Search Strategy). The search included studies published from database inception through 29 August 2025, with no restrictions on language. The protocol of this study was registered on PROSPERO (CRD420251133875).

### 2.3. Inclusion and Exclusion Criteria

Studies were considered eligible if they met the following criteria. (1) Observational studies, including case–control, cohort, and longitudinal designs, that investigated serum VitD levels, deficiency, or insufficiency in individuals with PD and HC were included. (2) Clinical trials assessing the efficacy of VitD supplementation compared to placebo in patients with PD were also eligible. Additionally, (3) case–control studies examining the association between vitamin D receptor (*VDR*) gene polymorphisms and susceptibility to PD were considered. (4) Only studies in which PD was clinically diagnosed by a neurologist according to the United Kingdom Parkinson’s Disease Brain Bank criteria were included. (5) Studies were required to report sufficient data on sample size, serum VitD levels, and genotypic distribution of the *VDR* gene. (6) All participants had to be adults aged 18 years or older.

(1) Studies were excluded if they were reviews, case reports, meta-analyses, editorials, or commentaries. (2) Studies lacking detailed information on genotyping, VitD status, or population characteristics were excluded. (3) Research focusing on genes other than *VDR* or reporting additional *VDR* genotypes not relevant to the analysis was also excluded. (4) Animal studies and studies that did not assess patients clinically diagnosed with PD were not considered.

No language restriction was applied. In cases where the non-English literature or when the English abstract provided insufficient information, as observed in articles written in Turkish, Chinese, Russian, the Google Translate service was employed [18].

### 2.4. Data Extraction

Two investigators (J.P.R. and A.L.F.C.) independently reviewed the literature and extracted relevant data. For each study, information was collected on the first author, country of origin, and year of publication. For studies comparing PD patients and HC, data were extracted on sample sizes, serum VitD levels, and the number of individuals classified as having VitD insufficiency or deficiency. In studies reporting incidence, the investigators recorded sample size, number of incident PD cases, VitD status, and HR. For clinical trials assessing VitD supplementation, data were extracted on sample size and motor outcomes, including UPDRS-Total, UPDRS-III, and Timed Up and Go (TUG) scores, all measured during the “ON” medication period. Any discrepancies between reviewers were resolved through discussion.

### 2.5. Quality Assessment

The Newcastle-Ottawa Scale (NOS) was used to evaluate the process in terms of case selection, comparability of queues, and evaluation of results. A study with a score of at least six was considered as high-quality literature. Higher NOS scores showed higher literature quality. The NOS was used to rate the quality of observational cohort and case–control studies and an adapted version of the NOS for quality of cross-sectional studies [19].

The risk of bias in randomized controlled trials (RCTs) was assessed using the ROBVIS tool, a web-based application designed to visually summarize risk of bias judgments based on established frameworks such as the Cochrane Risk of Bias tool [20].

### 2.6. Statistical Analysis

Statistical analyses were conducted to compare outcomes between individuals with PD and HC. Data extraction and organization were performed using Microsoft Excel (version 2023 for macOS). Serum VitD levels were reported as means with SD, and SMD were calculated using Cohen’s d to assess differences in VitD levels between PD and HC groups. For studies lacking reported SDs, values were estimated assuming a large effect size (Cohen’s d ≥ 0.8). Ninety-five percent confidence intervals (CIs) were derived using the standard error formula for Cohen’s d and the normal approximation. To ensure consistency with clinical standards, VitD concentrations originally reported in nanomoles per liter (nmol/L) were converted to nanograms per milliliter (ng/mL) using the molecular weight of the analyte.

To evaluate the association between VitD status (insufficiency and deficiency) and PD occurrence, ORs were calculated. For studies assessing the effect of VitD supplementation, SMDs were computed to compare motor outcomes between treatment and placebo groups, using pooled SDs and group means with sample sizes. For continuous outcomes, pooled SMDs were calculated using the inverse variance method, with between-study variance (τ^2^) estimated via the DerSimonian–Laird approach [21]. Confidence intervals for τ^2^ and τ were computed using the Jackson method. Heterogeneity was assessed using the I^2^ statistic derived from Cochran’s Q, and prediction intervals were calculated using a t-distribution with 2 degrees of freedom to estimate the expected range of effects in future studies. For binary outcomes, ORs with 95% CIs were pooled using both fixed-effect and random-effects models. The Mantel–Haenszel method was applied for fixed-effect analysis, while the inverse variance method was used for random-effects modeling, incorporating τ^2^ estimated via DerSimonian–Laird. The Mantel–Haenszel estimator was also used in the calculation of Q and τ^2^. Prediction intervals for binary outcomes were calculated using a t-distribution with 15 degrees of freedom. Heterogeneity was again assessed using I^2^, with values exceeding 50% indicating substantial variability [22].

To run all the above-mentioned analyses and demonstrate results via forest plots, we carried out initial calculations using Meta-Mar (version 4.1.2, https://www.meta-mar.com/, (accessed on 17 September 2025)), a free online meta-analysis service developed by Beheshti et al. [23]. Forest plots for *VDR* SNPs were generated using GraphPad Prism (version 10.4.1 for macOS; GraphPad Software, Boston, MA, USA; www.graphpad.com). For calculations of ORs and additive genetic models the analyses were performed using SPSS software (version 22.0; IBM Corp., Armonk, NY, USA). A *p*-value of less than 0.05 was considered statistically significant.

Genotypic data were analyzed to assess HWE in control groups using Fisher’s exact test for allele frequencies and Pearson’s chi-square test (df = 1). All polymorphisms were evaluated using the forward DNA strand to ensure data standardization. To investigate the association between *VDR* SNPs and PD, multiple genetic models were applied, including allele, dominant, recessive, and additive models. Comparisons were made using ORs, 95% CIs, and *p*-values. The allele model compared A (dominant) vs. a (recessive); the dominant model compared AA vs. Aa + aa; the recessive model compared aa vs. AA + Aa; and the additive model treated genotypes as ordinal variables (AA = 0, Aa = 1, aa = 2), assuming a linear increase in risk per recessive allele. Logistic regression was used to estimate additive effects, yielding ORs per allele with corresponding CIs and *p*-values. For studies lacking isolated allele counts, allele frequencies were calculated using the following formula: A alleles = (2 × AA) + (1 × Aa); a alleles = (2 × aa) + (1 × Aa) [24].

### 2.7. Definitions

According to the 2011 Endocrine Society Clinical Practice Guideline, serum 25(OH)D3 levels were categorized as follows: deficiency was defined as <20 ng/mL (50 nmol/L), insufficiency as 21–29 ng/mL (52.5–72.5 nmol/L), and sufficiency as ≥30 ng/mL (≥75 nmol/L) [25]. However, the updated 2024 guideline no longer endorses these thresholds for generally healthy individuals, citing insufficient evidence from randomized controlled trials to support specific cutoffs for clinical benefit [26].

Genetic analysis focused on several SNPs within the *VDR* gene and related loci (Appendix A–SNPs). The following polymorphisms were included: ApaI (rs7975232), A-1012G (rs4516035), BsmI (rs1544410), Cdx2 (rs11568820), FokI (rs2228570, also known as rs10735810), TaqI (rs731236), and Tru9I (rs757343). Additional SNPs analyzed included rs1989969, rs2853559, rs4334089, rs7299460, rs7968585, rs7976091, and rs10083198. Both the significant SNPs identified through GWAS and those incorporated into the optimal PRS model were annotated using the online tool SnpXplorer [27].

A variety of assays have been used to measure serum VitD levels in PD studies, each with distinct sensitivity and specificity profiles (Appendix A–Assays) [28,29,30,31,32,33,34,35]. Traditional methods like the Competitive Protein Binding Assay (CBPA) and Radioimmunoassay offer foundational approaches but are less commonly used today due to limitations in accuracy. More advanced techniques such as Liquid Chromatography–Mass Spectrometry (LC-MS) and LC-MS/MS provide high precision and are considered gold standards for quantifying VitD metabolites. Immunoassay-based methods, including ELISA, Electrochemiluminescence Immunoassay (ECL), and Chemiluminescent Microparticle Immunoassay, are widely used in clinical settings due to their efficiency and scalability, though they may vary in cross-reactivity and calibration standards. The choice of assay could have influenced reported VitD levels and should be considered when comparing results across studies.

## 3. Results

### 3.1. Study Selection

An initial screening of PubMed/Medline, Scopus, and Google Scholar yielded 2575 articles, of which 63 met the predefined inclusion criteria. A comprehensive summary of the study selection process is illustrated in the following Figure 2. There were 35 articles about VitD status [28,29,30,31,32,33,34,35,36,37,38,39,40,41,42,43,44,45,46,47,48,49,50,51,52,53,54,55,56,57,58,59,60,61,62], 8 clinical trials [63,64,65,66,67,68,69,70], and 20 assessing *VDR* SNPs [40,71,72,73,74,75,76,77,78,79,80,81,82,83,84,85,86,87,88,89]. A total of 192 analyses were performed across the studies.

### 3.2. Quality Assessment

The studies assessing VitD status had NOS scores ranging from 4 to 9, with a mean score of 7.26 and a median of 7.5 (Appendix A–NOS Levels) [28,29,30,31,32,33,34,35,36,37,38,39,40,41,42,43,44,45,46,47,48,49,50,51,52,53,54,55,56,57,58,59,60,61,62]. Of the 34 studies included, 26 were classified as high quality (NOS ≥ 7), 7 as moderate quality (NOS 5–6), and 1 as low quality (NOS < 5). Risk of bias was further evaluated using the ROBVIS tool, which indicated that 2 studies were rated as low risk, 5 had some concerns, and 1 was rated as high risk (Appendix A–Robvis) [63,64,65,66,67,68,69,70].

For the genetic polymorphism studies, NOS scores ranged from 5 to 8, with a mean of 6.40 and a median of 6.0. Among the 20 studies assessed, 6 were classified as high quality, 14 as moderate quality, and none were considered low quality (Appendix A–NOS Polymorphism) [40,71,72,73,74,75,76,77,78,79,80,81,82,83,84,85,86,87,88,89].

### 3.3. Meta-Analysis

#### 3.3.1. VitD Serum in PD

A meta-analysis of 33 studies involving 3920 individuals with PD and 6792 HC demonstrated a consistent pattern of lower serum VitD levels in PD patients (Figure 3) (Appendix A–Serum 25(OH)D) [28,29,30,31,32,33,34,35,36,37,38,40,41,42,43,44,45,46,47,48,50,51,52,53,54,55,56,57,58,60,61,62,86]. Employing the DerSimonian–Laird estimator and inverse variance weighting, the fixed effect model yielded an SMD of −0.46 (95% CI: −0.51 to −0.41), while the random effects model produced a slightly larger effect size of −0.59 (95% CI: −0.71 to −0.46). I^2^ value was 82%, Cochran’s Q test was significant (Q = 177.69, *p* < 2.2 × 10^−16^), suggesting notable variability among studies. Publication bias was assessed using multiple methods, all of which indicated significant small-study effects: Egger’s test (*p* = 0.00086), Begg’s test (*p* = 0.00157), and the Thompson-Sharp test (*p* = 1.76 × 10^−5^). The Trim-and-Fill method estimated 10 potentially missing studies, adjusting the effect size to −0.37 (95% CI: −0.51 to −0.24), representing a 36% reduction in magnitude.

#### 3.3.2. VitD Insufficiency in PD

A meta-analysis was conducted on 12 studies reporting binary outcomes, utilizing the metabin model and Odds Ratio (OR) as the summary measure (Figure 4) (Appendix A–25(OH)D Insufficiency) [29,30,31,33,34,35,42,44,48,49,53,58]. The analysis applied the DerSimonian–Laird estimator for between-study variance and employed the Mantel–Haenszel method for pooling. The fixed effect model produced a statistically significant OR of 1.52 (95% CI: 1.33 to 1.74), while the random effects model yielded an OR of 1.43 (95% CI: 1.05 to 1.96), accompanied by a wide prediction interval (0.51 to 4.05), indicating uncertainty in the estimate. I^2^ statistic was 75%, Cochran’s Q test was significant (Q = 43.65, *p* = 8.36 × 10^−6^), indicating notable variability among the studies. Assessment of publication bias through Egger’s test, Begg’s test, the Thompson-Sharp test, and the Trim-and-Fill method revealed no significant evidence of bias, with all *p*-values exceeding the 0.05 threshold and no missing studies identified. These findings indicate a statistically significant effect size with some heterogeneity, but no apparent publication bias, providing a reliable synthesis of the included studies.

#### 3.3.3. VitD Deficiency in PD

A meta-analysis included 17 studies investigating the association between VitD deficiency and PD compared to HC (Figure 5) (Appendix A–25(OH)D Deficiency) [29,31,32,34,35,39,41,42,43,44,47,48,49,51,53,58,59]. The fixed effect model yielded a statistically significant OR of 2.20 (95% CI: 1.91 to 2.53), while the random effects model produced a slightly higher OR of 2.38 (95% CI: 1.79 to 3.17), with a wide prediction interval (0.85 to 6.66), indicating variability across studies. Heterogeneity assessment revealed a τ^2^ of 0.21 and an I^2^ of 69%, Cochran’s Q statistic (Q = 51.35, *p* = 1.395 × 10^−5^) indicated significant variability. Publication bias analysis showed mixed results: Egger’s test was non-significant (*p* = 0.17), while Begg’s test and the Thompson-Sharp test indicated significant bias (*p* = 0.03 and 0.02, respectively). The Trim-and-Fill method estimated five potentially missing studies, adjusting the effect size from 0.86 to 0.63, reflecting a 27.2% reduction and suggesting the presence of publication bias.

#### 3.3.4. VitD Serum and PD Incidence

The comparative analysis of two large cohort studies investigated the relationship between VitD status and the risk of developing PD (Appendix A–PD Incidence) [90,91]. Gröninger et al. reported a HR of 1.08 (95% CI: 0.80–1.45) for individuals with VitD deficiency, suggesting a slightly increased but statistically non-significant risk [90]. In contrast, Veronese et al. found a significantly reduced risk of PD among individuals with sufficient VitD levels, with an HR of 0.83 (95% CI: 0.74–0.93) [91]. These findings imply that maintaining adequate VitD levels may offer a protective effect against PD, while deficiency does not show a strong or conclusive association with increased risk. The standardized incidence rates further support this trend, with higher PD occurrence observed in the sufficient group (662.31 vs. 328.56 per 100,000 people), likely reflecting differences in sample size and follow-up duration rather than contradicting the protective association.

#### 3.3.5. VitD Supplementation in PD

The effects of VitD supplementation versus placebo on motor outcomes in patients with PD were assessed across eight studies, comprising a total of 284 individuals with PD and 282 HC (Figure 6) (Appendix A–VitD Supplementation) [63,64,65,66,67,68,69,70]. For UPDRS Total scores during “ON” periods, the fixed effect model yielded an SMD of −0.08 (95% CI: −0.30 to 0.14), while the random effects model produced a slightly larger effect size of −0.39 (95% CI: −1.10 to 0.33). The I^2^ value was 89%, and Cochran’s Q test was significant (Q = 27.52, *p* < 4.57 × 10^−6^). For UPDRS-III scores, the fixed effect model yielded an SMD of −0.10 (95% CI: −0.27 to 0.07), while the random effects model produced a slightly larger effect size of −0.18 (95% CI: −0.49 to 0.12). The I^2^ value was 63%, and Cochran’s Q test was significant (Q = 16.41, *p* < 1.17 × 10^−2^). Regarding TUG performance, the fixed effect model yielded an SMD of −0.18 (95% CI: −0.43 to 0.06), while the random effects model produced a slightly larger effect size of −0.17 (95% CI: −0.44 to 0.10). The I^2^ value was 8%, and Cochran’s Q test was significant (Q = 3.27, *p* < 3.51 × 10^−1^). These findings suggest that VitD supplementation may have modest and variable effects on motor function in PD, with the most consistent trend observed in TUG performance.

#### 3.3.6. VitD Polymorphism

Several SNPs showed varying degrees of association with PD (Figure 7) (Appendix A–Polymorphism) [40,71,72,73,74,75,76,77,78,79,80,81,82,83,84,85,86,87,88,89]. ApaI (rs7975232) was widely studied, with some significant findings in allele and additive models, though overall results were inconsistent. BsmI (rs1544410) also showed sporadic significance, particularly in the additive and recessive models in studies by Kim et al. [71] and Gatto et al. [92]. In contrast, SNPs like A-1012G (rs4516035), BglI (rs739837), and Cdx2 (rs11568820) consistently showed no significant associations across all genetic models.

The most robust and consistent associations were observed with FokI (rs2228570). Multiple studies, including Han et al. [72], Török et al. [75], Hu et al. [84], and Agliardi et al. [85], reported significant results in allele and additive models, with overall meta-analytic significance (e.g., OR = 1.15, *p* = 0.03 in recessive model). TaqI (rs731236) showed mixed results, with Gatto et al. reporting significant associations in allele and additive models [92], while other studies did not replicate these findings. rs4334089 also showed significance in specific studies, particularly in allele and additive models reported by Fazeli et al [78].

Other SNPs such as Tru9I (rs757343), rs1989969, rs2853559, rs7299460, and rs7968585 generally lacked significant associations. However, rs7976091 showed a notable additive effect in Kang et al. (OR = 2.24, *p* < 0.01) [79], and rs10083198 demonstrated a strong additive association (OR = 2.05, *p* < 0.01) in Lin et al.’s study [76].

The HWE results show that several SNPs, including ApaI (rs7975232), FokI (rs2228570), and TaqI (rs731236), exhibited significant deviations from HWE in multiple studies, particularly in case groups, suggesting possible genetic disequilibrium or sampling biases. Conversely, SNPs such as A-1012G (rs4516035), BglI (rs739837), and Cdx2 (rs11568820) generally maintained equilibrium across studies, with *p*-values well above 0.05.

## 4. Discussion

### 4.1. Summary of Results

The current systematic review and meta-analysis of 63 studies investigated the relationship between VitD and PD, including 35 on VitD status, 8 clinical trials, and 20 on *VDR* SNPs. Quality assessment revealed that most studies were of moderate to high quality, with VitD studies averaging a NOS score of 7.26 and genetic studies averaging 6.40. Meta-analyses showed that PD patients had significantly lower serum VitD levels (SMD = −0.46), higher odds of VitD insufficiency (OR = 1.52), and deficiency (OR = 2.20), with high heterogeneity across studies. Cohort data suggested that sufficient VitD levels may reduce PD risk (HR = 0.83), while deficiency showed no strong association. VitD supplementation had modest effects on motor outcomes, with the most consistent improvement seen in TUG performance. Genetic analyses identified FokI (rs2228570) as the most consistently associated SNP with PD, while ApaI and BsmI showed variable significance. Other SNPs like A-1012G, BglI, and Cdx2 showed no consistent associations. HWE analysis revealed deviations in SNPs such as ApaI, FokI, and TaqI, particularly in case groups, suggesting potential genetic imbalance or sampling bias.

### 4.2. Motor and Non-Motor PD Symptoms

There have been increasing reports linking both motor and non-motor symptoms of PD with serum VitD levels (Table 1) [45,48,49,67,91,93,94,95,96,97,98,99,100,101,102,103,104]. In this chapter, we will review selected studies that explore these associations, highlighting the potential role of VitD in the clinical presentation and progression of PD.

Iranian patients with PD showed significantly lower serum 25(OH)D3 levels compared to normal ranges. This deficiency was notably linked to increased postural instability, abnormal posture, and freezing episodes [95]. Similarly, an Egyptian study found that VitD levels were negatively correlated with age and age at onset; they were not significantly associated with disease duration or severity (UPDRS and H&Y) [49]. Zhang et al. reported that lower VitD levels were significantly associated with increased falls, insomnia, higher scores for depression and anxiety, and poor sleep quality, but not with UPDRS scores or disease duration [48]. Similarly, Meamar et al. found that serum VitD levels were negatively correlated with UPDRS scores (r = −0.34, *p* < 0.05) [102]. Peterson et al. showed that VitD levels were correlated with motor symptom severity and gait features, such as automatic postural response strength and stance weight asymmetry during backward perturbation tasks [96].

In the oxford discovery cohort of early PD patients, serum VitD levels averaged 49.1 nmol/L (SD 25.0), with lower levels significantly associated with worse baseline activities of daily living (UPDRS-II intercept: −0.75, *p* = 0.005, q = 0.007). However, VitD did not predict motor or cognitive progression over time (UPDRS-III slope: *p* = 0.91, q = 0.91) [103]. Similarly, Chitsaz et al. found a high prevalence of VitD insufficiency (72.8%) and deficiency (38.4%) among Iranian PD patients, but no significant association between serum 25(OH)D3 levels and disease severity as measured by H&Y or UPDRS-III scores [105].

Kim et al. showed that VitD levels were independently associated with olfactory dysfunction and NMSS, with stronger associations observed in the akinetic-rigid subtype [101]. Kwon et al. revealed that low serum VitD levels were independently correlated with delayed gastric emptying in drug-naive, de novo PD patients [100]. Also, Canlı et al. found a strong inverse correlation between fatigue severity and both VitD concentration (r = −0.851, *p* < 0.001) and physical activity (r = −0.757, *p* < 0.001), while disease stage (r = 0.879, *p* < 0.001) and anxiety levels (r = 0.797, *p* < 0.001) were positively associated with fatigue intensity [98].

Jang et al. found that PD patients with OH had significantly reduced levels of both 25(OH)D3 and calcitriol compared to those without OH, and these levels were negatively correlated with blood pressure changes and symptom severity [99]. However, Arici Duz et al. showed no significant differences in VitD levels among dipper, non-dipper, and reverse dipper (*p* = 0.192), suggesting that VitD status was not associated with autonomic dysfunction [106].

Peterson et al. demonstrated that higher serum VitD concentrations were significantly associated with better verbal fluency and memory (e.g., vegetable fluency: β = 0.264, t = 4.31, *p* < 0.001; HVLT immediate recall: β = 0.196, t = 3.04, *p* = 0.0083) and lower depression scores (GDS: β = −0.205, t = −3.08, *p* = 0.0083) in non-demented PD patients [97]. However, Lien et al. found no direct correlation between serum VitD levels and cognitive or motor severity in PD. Additionally, PD patients at risk of malnutrition showed worse UPDRS scores (*p* = 0.046) and poorer memory and calculation performance (*p* = 0.002 and *p* = 0.010) [107]. Expanding on this, Barichella et al. demonstrated that lower serum 25(OH)D3 levels (mean 17.4 ng/mL) were independently associated with more severe motor symptoms (UPDRS Part III: β = −2.10, *p* = 0.006), higher H&Y stage (β = −0.08, *p* = 0.035), and poorer global cognition (MMSE: β = 0.47, *p* = 0.041), despite similar VitD intake (2.5 μg/day) and sunlight exposure across groups [51].

In a longitudinal study of 60 untreated, de novo PD patients, 93.3% had VitD insufficiency at baseline, with a mean serum 25(OH)D3 level of 17.6 ± 7.3 ng/mL. After 48 months, patients who developed mild cognitive impairment (PD-MCI) had significantly lower baseline VitD levels (12.7 ± 5.4 ng/mL) compared to those who remained cognitively normal (19.7 ± 7.8 ng/mL, *p* = 0.005), and logistic regression confirmed low 25(OH)D3 as an independent predictor of PD-MCI (OR = 0.854, 95% CI: 0.746–0.977, *p* = 0.022) [108].

Resting-state fMRI revealed that patients with lower VitD levels exhibited altered spontaneous neuronal activity in default-mode and visual networks, suggesting a dose-dependent impact of VitD on brain function [104]. In this context, Russel et al. observed a progressive decline in serum 25(OH)D3 levels and regional cerebral blood flow in the basal ganglia as clinical stages advanced. Notably, patients in stages 4 and 5 exhibited significantly lower HMPAO uptake ratios and VitD concentrations compared to those in earlier stages [109].

Ogura et al. demonstrated that VitD biomarkers can effectively differentiate patient groups, with 25(OH)D3 showing strong accuracy in identifying both PD and MSA from HCs, while 1,25(OH)_2_D was more effective in distinguishing MSA from PD [50]. Supporting this, Guo et al. [93] found that serum levels of 25(OH)D3, along with Klotho and homocysteine, were significantly altered in MSA patients compared to healthy controls, and that these biomarkers correlated with disease severity, including motor and cognitive impairments. Although PD patients also exhibited changes in these biomarkers, the combination of Klotho, 25(OH)D3, and homocysteine provided superior diagnostic accuracy for distinguishing MSA from PD, particularly in male patients [93]. In patients with PD, CSF levels of VitD binding protein (VDBP) were significantly elevated compared to HCs, suggesting a potential involvement of VitD transport or regulation in PD pathology. This increase in VDBP was among the top contributors to distinguishing PD cases in a multianalyte biomarker profile, highlighting its relevance in disease classification [110].

Thaler et al. found that serum VitD levels were not significantly different across PD subtypes (idiopathic, GBA-related, and LRRK2-related). However, among non-manifesting LRRK2 mutation carriers, lower VitD levels were significantly associated with higher prodromal PD probability scores (β = −0.615, 95% CI: −1.179 to −0.051, *p* < 0.001) [94]. Tassorelli et al. reported that PD patients with a history of fall-related fractures had significantly lower VitD levels (8.5 ± 5.3 ng/mL) compared to those without fractures (16.9 ± 15.7 ng/mL, *p* = 0.01) [111]. Deng et al. found no statistically significant differences in serum VitD levels among the three PD subtypes. The mean concentrations were 22.8 ± 7.5 ng/mL in the severe cluster (A), 23.2 ± 7.3 ng/mL in the intermediate cluster (B), and 21.7 ± 6.7 ng/mL in the mild cluster (C), with a *p*-value 0.52 [112].

In patients younger than 60 years, serum VitD levels were significantly lower in the PD group (20.24 ± 12.62 ng/mL) compared to HCs (32.46 ± 19.14 ng/mL; *p* = 0.010). This deficiency was associated with increased disease severity and duration, suggesting that lower VitD levels may contribute to earlier and more aggressive progression of PD in younger individuals [45]. Similarly, in patients aged 60 and older, serum VitD levels were significantly lower in the PD group (18.40 ± 5.91 ng/mL) compared to controls (28.13 ± 6.26 ng/mL; *p* = 0.001). Among older patients, those with higher disease severity (scores above 3) had the lowest VitD levels, particularly males (11.80 ± 2.12 ng/mL). Furthermore, VitD levels declined with longer disease duration, especially in females, where those with over 10 years of PD had levels of 23.40 ± 4.67 ng/mL, showing a statistically significant trend (*p* = 0.049) [61].

A three-month regimen combining butyrate triglyceride, Crocus sativus L., and VitD3 led to a notable 7.7% improvement in motor function and increased stool frequency among PD patients. This formulation, acting through the gut–brain axis, demonstrated potential as a complementary therapy targeting both motor and non-motor symptoms, particularly constipation [113].

### 4.3. Outdoors Sunlight Exposure

Huang et al. reported that outdoor time during summer is associated with a decreased risk of PD, and this was statistically more significant in individuals with genetic mutations characterized by high penetrance [114]. Noteworthy, some aspects like physical activity, sleep patterns, and VitD may likely be confounding factors for these findings. Supporting this, Hu et al. found that spending more than 3.5 h outdoors daily was associated with a 15% reduction in PD risk (HR = 0.85, 95% CI: 0.75–0.96) and that increased outdoor time was positively correlated with serum VitD levels, which themselves were linked to a lower PD risk [115]. Complementing these findings, Kenborg et al. conducted a large population-based case–control study in Denmark involving 3819 male PD patients and 19,282 controls, revealing that maximal outdoor work was associated with a 28% lower risk of PD (adjusted OR = 0.72, 95% CI: 0.63–0.82) [116]. Extending this evidence, Kravietz et al. performed a nationwide ecological study in France using ultraviolet B (UV-B) radiation as a proxy for VitD exposure. Analyzing over 69,000 incident PD cases, they found a quadratic, age-dependent association between UV-B exposure and PD incidence: in individuals under 70, higher UV-B exposure was linked to reduced PD risk (e.g., RR = 0.85, 95% CI: 0.77–0.94 for ages 45–49), whereas in older adults, the trend reversed [117].

Zhu et al. found that individuals engaging in more than 6 h of outdoor activity per week had a 56% lower risk of PD (adjusted OR = 0.437, 95% CI: 0.241–0.795), and participants in the highest quartile of total VitD intake (>12 μg/day) had a 46% reduced risk (adjusted OR = 0.538, 95% CI: 0.301–0.960) [118]. Similarly, Kwon et al. conducted a population-based case–control study in western Washington State and observed a modest inverse association between outdoor occupational exposure and PD risk. Specifically, individuals whose entire job history involved outdoor work had a lower risk of PD compared to those with exclusively indoor work (OR = 0.74, 95% CI: 0.44–1.25), although the trend was not statistically significant [119].

### 4.4. In Vitro and In Vivo Studies with VitD and PD

In vitro and in vivo studies have increasingly explored the role of VitD in the pathophysiology of PD, revealing its potential neuroprotective and neuromodulatory effects (Figure 8) (Appendix A–In Vitro) [7,8,120,121,122,123,124,125,126,127,128,129,130,131,132,133,134,135,136] (Appendix A–In Vivo) [8,130,132,134,137,138,139,140,141,142,143,144,145,146,147,148,149,150,151,152,153,154,155,156,157,158,159,160,161,162] (Table 2) [133,144,163,164,165,166,167,168].

#### 4.4.1. In Vitro

Gul et al. investigated the neuroprotective effects of 1,25(OH)2D3 (calcitriol) and a multi-strain probiotic formulation (SLAB51) against rotenone-induced neurotoxicity in SH-SY5Y human neuroblastoma cells, a model for PD [169]. Rotenone (150 nM) significantly reduced cell viability, mimicking PD-like oxidative stress. Pretreatment and post-treatment with calcitriol (1.25–5 µM) and probiotics (0.01–0.1 mg/mL), alone or in combination, significantly improved cell viability and modulated antioxidant enzyme levels. Specifically, treatments increased SOD, GSH, and CAT activities while reducing GSR levels. Calcitriol at 2.5 µM and probiotics at 0.05–0.1 mg/mL were particularly effective. The combination therapy showed synergistic effects, with the highest cell viability increase (66.4%) observed in post-treatment groups.

Rotenone, a mitochondrial complex I inhibitor, induces excessive ROS production, leading to oxidative stress and dopaminergic neurodegeneration in Parkinson’s disease models. Antioxidant enzymes such as SOD, catalase, and glutathione peroxidase play a pivotal role in neutralizing ROS and maintaining redox homeostasis. Experimental studies in rotenone-induced PD models have shown that enhancing these enzyme activities significantly reduces oxidative markers like thiobarbituric acid reactive substances (TBARS), restores GSH levels, and improves neuronal survival and motor function [170]. Compounds such as quercetin and selenium have been reported to upregulate antioxidant enzyme expression, mitigating rotenone-induced neurotoxicity and apoptosis, highlighting their therapeutic potential in PD management [171].

VDR plays a crucial neuroprotective role in PD by reducing oxidative stress in dopaminergic neurons and suppressing inflammation in microglia via inhibition of the NLRP3/caspase-1 pathway [13]. In both cellular and *Caenorhabditis elegans* PD models, VDR activation—either through vitamin D3 supplementation or genetic upregulation—restores mitochondrial function, reduces αSyn toxicity, and improves motor behavior. Importantly, Zheng et al. identified DUB3 as a novel deubiquitinase that stabilizes VDR protein levels by interacting with its ligand-binding domain, thereby enhancing VDR’s protective effects and offering a promising therapeutic target for PD intervention.

Pro-inflammatory and pro-apoptotic pathways are central to PD-related neurodegeneration, as chronic neuroinflammation amplifies oxidative stress and triggers caspase-mediated neuronal apoptosis. VitD’s immunomodulatory properties are particularly relevant here, as it downregulates pro-inflammatory cytokines (e.g., TNF-α, IL-6) and inhibits NF-κB signaling, thereby reducing microglial activation and subsequent neuronal damage [172]. This mechanism underscores its potential neuroprotective role in mitigating both inflammatory and apoptotic cascades that accelerate dopaminergic neuron loss. Furthermore, previous studies have demonstrated that VitD, its receptor (VDR), and activating enzymes (such as 1α-hydroxylase and CYP27B1) are expressed in the substantia nigra [173]. Low VitD levels may therefore impair VDR-mediated signaling, contributing to dysfunction or death of dopaminergic neurons. Such deficits could exacerbate motor symptoms like bradykinesia and rigidity, as well as non-motor features including cognitive decline and mood disturbances [174].

NSC-34 motor neuron-like cells express key enzymes involved in VitD metabolism, including CYP24A1, and actively convert 25(OH)D3 into 24,25OHD3. The expression of CYP24A1 is upregulated by calcitriol and synthetic analogs, indicating a functional regulatory mechanism. These findings suggest NSC-34 cells are a promising in vitro model for investigating neuronal VitD metabolism [175]. Compared to in vivo models like 6-OHDA-lesioned rats or MPTP-treated mice, NSC-34 cells offer a controlled environment to study molecular mechanisms without systemic variables. Unlike whole-animal models, they allow direct manipulation of VitD pathways at the cellular level.

VitD significantly downregulated LAMP3 expression in human monocyte-derived dendritic cells, reducing mRNA levels by 1.62-fold at 24 h and 2.2-fold at 168 h during differentiation (*p* < 0.001). This suppression, mediated via NF-κB pathway inhibition, mirrors the tolerogenic phenotype induced by IL-10 and dexamethasone, suggesting a potential immunomodulatory role relevant to PD, where LAMP3 is a known genetic risk factor [176].

#### 4.4.2. In Vivo

In a rotenone-induced PD model using Wistar rats, VitD was administered orally at 500 IU/kg/day, either as monotherapy or combined with canagliflozin (10 mg/kg/day) [177]. VitD alone improved striatal dopamine content, reduced oxidative stress markers such as malondialdehyde (MDA), and increased antioxidant defenses including GSH and catalase (CAT). It also attenuated inflammatory cytokines (TNF-α, IL-6, IL-10) and partially normalized NF-κB and Nrf2 expression. Histological analysis revealed preservation of striatal architecture and reduced neuronal damage, while immunohistochemistry showed improved tyrosine hydroxylase (TH) expression and decreased α-synuclein accumulation. Although the combination therapy produced more pronounced effects across all parameters, the study design does not allow clear attribution of these benefits to VitD alone. Therefore, only monotherapy findings should be considered for this review to avoid confounding effects from canagliflozin.

Experimental models, such as MPTP- and 6-OHDA-induced PD in rodents, have shown that VitD supplementation significantly reduces oxidative stress and inflammation, preserves dopaminergic neurons in the substantia nigra, and improves motor coordination [152,154]. These effects are mediated through the upregulation of anti-inflammatory cytokines (e.g., IL-10, TGF-β) and downregulation of pro-inflammatory markers (e.g., TNF-α, IL-1β) [178], as well as through enhanced expression of neurotrophins like BDNF [179] and GDNF [133]. Additionally, VitD modulates neurotransmitter systems, particularly the cholinergic and dopaminergic pathways, by restoring acetylcholine levels and reducing acetylcholinesterase activity [180].

Emerging evidence also suggests that VitD may regulate iron homeostasis and inhibit ferroptosis by modulating the expression of iron transport and detoxification proteins [181]. Furthermore, VitD influences synaptic plasticity and ion channel function, including L-type calcium channels [182] and NMDA receptors [183]. Epigenetically, VitD acts through *VDR* to modulate gene expression via histone modification and DNA methylation [184].

Calcitriol significantly improved motor performance and protected dopaminergic neurons in 6-OHDA-induced hemiparkinsonian mice [185]; it promoted Treg expansion, which suppressed T-cell infiltration and microglial M1 polarization, reducing pro-inflammatory cytokines like TNF-α, IL-1β, and IL-6; depletion of Treg using anti-CD25 antibody abolished calcitriol’s anti-inflammatory effects.

VitD significantly attenuated L-DOPA-induced abnormal involuntary movements (AIMs) in a mouse model of PD, showing a maximal reduction of 32.7% on day 14 of treatment without altering dopamine metabolic enzymes such as TH and MAO-B. Additionally, VD3 reduced oxidative stress markers like p47phox and inflammatory cytokines such as IL-1β [186].

In a haloperidol-induced mouse model of Parkinsonism, supplementation with VitD3 (800 IU) and vitamin A (1000 IU) for 21 days significantly improved motor function, as evidenced by increased bilateral climbing attempts and reduced unsuccessful climbing attempts compared to the haloperidol-only group. Additionally, the combination of VitD and A markedly reduced MDA levels, indicating decreased lipid peroxidation, and restored SOD activity to near-control levels, suggesting enhanced antioxidant defense in the striatum and primary motor cortex [187].

Vitamin D3 supplementation (800 IU/day) in dopamine-deficient mice (-D2) + VD group) partially reduced neurodegenerative changes in the visual cortex (*p* < 0.05) and improved retinal structure compared to the untreated -VitD2 group, although LDH levels remained elevated, indicating ongoing cytotoxic stress. Notably, when VitD was combined with vitamin A (-D2 + VD + VA group), oxidative stress markers (MDA) and cytotoxicity (LDH) were significantly lower, and histological analysis showed complete preservation of retinal architecture and absence of cortical degeneration, highlighting a synergistic protective effect [188].

In a mouse model selectively bred for high voluntary wheel-running, increased expression of the *VDR* was observed in both the cerebellum and striatum, suggesting a role in locomotor regulation and reward-dependent behavior. Zhang et al. showed a log2 fold change of −3.37 in cerebellum and −0.39 in striatum for *VDR* expression between control and high-activity lines (FDR-adjusted *p* = 4.4 × 10^−4^) [189].

#### 4.4.3. Computational Modeling

Computational screening identified three NURR1 agonists—Lifechemical_16901310, Maybridge_2815310, and NPACT_392450—with strong binding affinities (−59.06 to −46.77 kcal/mol) and stable molecular dynamics profiles, suggesting that VitD-related pathways may be leveraged for neuroprotection. These compounds exhibited favorable ADME properties and high reactivity (HOMO-LUMO gaps of −0.14 to −0.18 eV) [190].

#### 4.4.4. Human Studies

A 12-week BMI-adjusted VitD supplementation in PD patients with deep brain stimulation resulted in measurable biochemical changes [191]. The intervention group showed a reduction in TNF-α levels (from 7.07 ± 2.30 to 5.98 ± 2.09 pg/mL), while the placebo group experienced an increase (from 6.30 ± 2.00 to 7.23 ± 1.94 pg/mL). Additionally, 3-hydroxykynurenine (3-HK), a neurotoxic metabolite, remained stable in the VitD group but increased in the placebo group, with post-treatment levels of 6.23 ± 1.73 ng/mL versus 8.78 ± 4.93 ng/mL. Picolinic acid, known for its neuroprotective role, decreased in the placebo group but was preserved in those receiving VitD. These findings suggest that VitD may help regulate inflammation and neurotoxicity in PD.

Patients with PD showed reduced serum levels of VitD metabolites, with CYP27A1 significantly lower in PD and CYP2R1 and CYP24A1 levels remained unchanged [52]. Mazzetti et al. found that CYP27B1, the VitD–activating enzyme, was upregulated in a subpopulation of astrocytes in PD brains, particularly in regions affected by pathology, and these astrocytes were associated with αSyn oligomer clearance and neuroprotection [173]. Alaylıoğlu et al. demonstrated that although DHCR7/NADSYN1 and CYP2R1 gene variants were not associated with serum 25(OH)D3 levels or PD risk, specific SNPs in these loci correlated with milder motor symptoms, shorter disease duration, and reduced incidence of freezing of gait and falls [192].

Patients with PD and small intestinal bacterial overgrowth showed significantly lower serum triglyceride (72.8 ± 12.7 mg/dL vs. 100.2 ± 28.6 mg/dL, *p* = 0.024) and total bilirubin levels (0.86 ± 0.26 mg/dL vs. 0.98 ± 0.21 mg/dL, *p* = 0.019), suggesting impaired bile acid metabolism [193]. Hasuike et al. hypothesize that reduced bile acid from dysbiosis may hinder VitD absorption, contributing to osteoporosis and fractures commonly observed in PD.

### 4.5. VitD Polymorphisms and PD

*VDR* signaling plays a pivotal role in regulating gene expression involved in neurogenesis. Upon activation by 1,25(OH)2D3 (calcitriol), *VDR* forms a heterodimer with the RXR, which binds to VitD response elements in target gene promoters. This complex recruits co-activators such as DRIP and SRC, facilitating transcriptional activation and chromatin remodeling [194]. *VDR* signaling intersects with key developmental pathways like Wnt/β-catenin and SHH [195], which are essential for NSC proliferation and differentiation.

There are over 45 unique mutations in the *VDR* gene, including SNPs such as FokI (rs2228570), BsmI (rs1544410), ApaI (rs7975232), and TaqI (rs731236), which have been linked to altered susceptibility to PD. Mutations in the DNA-binding domain (e.g., Arg30) and ligand-binding domain (e.g., Arg274, Arg391) impair *VDR’s* ability to bind DNA, interact with co-activators, or heterodimerize with RXR, thereby disrupting downstream signaling [196]. For example, the Arg274Leu mutation reduces calcitriol binding affinity, while Glu420Lys affects co-activator recruitment without altering DNA or ligand binding [197].

Gholipour et al. showed that *VDR* was significantly overexpressed in PD patients, especially among females, while SNHG6 was notably underexpressed, particularly in female patients. No significant differences were found for SNHG16 and LINC00346. *VDR* and SNHG6 expression levels demonstrated diagnostic potential in PD with AUC values of 0.86 and 0.66, respectively [198].

Gatto et al. identified a significant association between the FokI recessive allele of the *VDR* gene and cognitive decline in PD, with each additional copy of the allele linked to a 0.115-point annual decrease in MMSE scores (β = −0.115, SE(β) = 0.05, *p* = 0.03). This effect was comparable in magnitude to the impact of disease duration and education level on cognitive performance [92].

VPA modulates the *VDR* pathway by altering the expression of nine key proteins involved in chromatin remodeling and transcriptional regulation, including HDAC1, SMARCA4, SMARCC1, SMARCD1, SMARCE1, BAZ1B, NCOA2, SUPT16H, and TOP2B. At a concentration of 2 mM, VPA downregulated these proteins in SH-SY5Y neuroblastoma cells, suggesting that VPA may influence VitD-mediated gene expression through epigenetic mechanisms such as histone acetylation and chromatin accessibility [199].

Genetic variants in the CACNA1C gene significantly influence PD risk, but only in individuals with VitD deficiency, suggesting a gene-environment interaction. Specifically, the G allele of SNP rs34621387 was associated with PD in VitD–deficient individuals (OR = 2.0–2.1, *p* = 0.002), while no association was found in non-deficient individuals (*p* > 0.8); additionally, VitD deficiency itself was linked to increased PD risk in both data sets (UDALL OR = 2.64, NGRC OR = 1.56) [200].

Butler et al. conducted a comprehensive genetic analysis of the *VDR* gene to assess its role in PD, focusing on both disease risk and age-at-onset (AAO). Using a two-stage design, they first genotyped 30 tagSNPs in 770 non-Hispanic Caucasian PD families and found that several SNPs in the 5′ region of *VDR*—particularly rs4334089—were significantly associated with earlier AAO (*p* = 0.0008). These SNPs were located in intronic regions between alternatively spliced exons, suggesting a possible mechanism involving transcript variation. In the second stage, they validated these findings using GWAS data from the NINDS cohort, where SNP rs7968585 in the 3′ region showed a significant association with AAO (*p* = 0.003), though not with disease risk [201].

### 4.6. Dietary Intake of VitD and PD

NHANES data (2007–2016) suggests that dietary intake of VitD does not show a statistically significant association with PD. Although PD patients consumed slightly less VitD (mean intake: 4.27 µg) compared to non-PD individuals (4.80 µg), the difference was modest and the adjusted OR (OR = 0.98) did not reach statistical significance (*p* = 0.17) [202]. Also, an MR analysis using 104 SNPs also showed no causal relationship between genetically predicted 25(OH)D3 levels and PD risk (IVW OR = 1.08; 95% CI: 0.90–1.29; *p* = 0.39), confirming the observational findings [62]. Yong et al. demonstrated a U-shaped association between serum 25(OH)D3 levels and all-cause mortality in PD patients, with the lowest risk observed at 78.68 nmol/L, and significantly increased mortality at both <50 nmol/L (HR 3.52, 95% CI 1.58–7.86) and >100 nmol/L (HR 2.92, 95% CI 1.06–8.05) [203].

Turcu-Stiolica et al. investigated the effects of daily supplementation with VitD3 (800 IU), folic acid (1000 µg), and vitamin B12 (15 µg) over six months in 24 levodopa-treated PD patients. The intervention significantly reduced serum homocysteine levels (from 19.98 ± 4.99 to 16.18 ± 3.73 µmol/L, *p* < 0.0001) and increased VitD levels (from 18.96 ± 7.28 to 22.68 ± 7.65 ng/mL, *p* = 0.025), while vitamin B12 levels remained statistically unchanged. Importantly, patients reported a significant improvement in health-related QoL, particularly in dimensions such as speech, mental function, discomfort and symptoms, and depression, with the overall 15D score increasing from 0.75 ± 0.11 (*p* = 0.02) [204].

In a cohort of 157 patients with early PD from the DATATOP study, 69.4% had VitD insufficiency (25[OH]D < 30 ng/mL) and 26.1% had deficiency (<20 ng/mL) at baseline. Over an average follow-up of 18.9 months, mean serum 25[OH]D levels increased from 26.3 ng/mL to 31.3 ng/mL, suggesting that VitD status does not decline during early disease progression [205].

In a cross-sectional study of Belgian PD patients, 55.9% had an inadequate intake of VitD, with mean daily intake at 10.1 ± 3.9 µg, below the recommended 10 µg threshold for many participants. Despite 64.4% of patients being aware of the interaction between dietary proteins and levodopa, only 18.2% consistently took their medication outside of meals, highlighting a gap in practical application of nutritional knowledge [206].

In a cross-sectional survey of 205 individuals with PD, 83.4% reported using at least one dietary supplement in the past 30 days, with VitD (74.3%), vitamin B12 (56.1%), and multivitamins (52.6%) being the most common. Notably, 43% of users took six or more supplements daily, and 29.2% had not discussed their supplement use with a healthcare provider [207].

### 4.7. Amount of VITD and Recommendations

The 2024 Endocrine Society guideline recommends VitD supplementation tailored to age, deficiency status, and comorbidities. For general adults up to age 74, routine supplementation is not broadly advised, though doses of 600–800 IU/day are suggested depending on age. Deficient individuals (<29 ng/mL) should receive 1000–2000 IU/day or 50,000 IU/week for 6–8 weeks, aiming for serum levels of 30–40 ng/mL. Obese and deficient individuals require higher doses (up to 100,000 IU/week). Those with diabetes or pre-diabetes should take 3500 IU/day. Adults aged 75+ may benefit from empiric supplementation (~900 IU/day), with similar deficiency protocols. Vitamin D3 is preferred, routine testing is discouraged, and calcium co-supplementation is advised for older adults (Table 3) [208].

Older adults and those with PD should maintain sufficient VitD levels to support bone and brain health. VitD levels should be monitored through blood tests, and supplementation should be considered if levels are low, under the guidance of a healthcare provider. Safe sun exposure—5–15 min, 2–3 times per week—is also advised to naturally boost VitD. Excessive supplementation should be avoided due to risks of toxicity, and individuals taking medications should consult their providers about potential interactions [209].

While the body can produce VitD through sunlight exposure, dietary sources are essential—especially for individuals with limited sun exposure or increased needs. Fatty fish such as rainbow trout, salmon, and tuna are among the richest natural sources, offering substantial amounts per serving. Fortified foods like milk, plant-based beverages (soy, almond, rice), yogurt, and orange juice also contribute meaningfully to daily intake. Additionally, UV-exposed mushrooms provide a plant-based option with variable but potentially high VitD content (Figure 9) (Appendix A–Daily VitD) [209].

VitD supplementation plays a crucial role in maintaining neurological health and may offer protective effects against PD. However, excessive intake can be harmful, as Chatterjee et al. reported a case of reversible parkinsonism induced by VitD toxicity, where a patient developed tremors, rigidity, and cognitive impairment after prolonged high-dose supplementation (60,000 IU/week for four years), resulting in hypercalcemia and suppressed parathyroid hormone levels [210]. Similarly, in klotho-deficient mice, elevated VitD activity triggered age-related loss of dopaminergic neurons in key brain regions, a process that was fully reversed by dietary VitD restriction [211]. This underscores the importance of balanced supplementation and medical oversight, as over-the-counter misuse of VitD can mimic or exacerbate neurodegenerative symptoms.

### 4.8. Conflicting Results

Kuhn et al. conducted a case–control study with 60 PD patients and matched controls, reporting mean 25(OH)D3 levels of 17.59 ± 9.28 µg/L in PD patients versus 15.69 ± 8.43 µg/L in controls, with no significant difference or correlation to disease severity or cognitive scores [212]. Fullard et al., using the PARS cohort of 198 asymptomatic individuals, found that those at high risk for PD had slightly higher vitamin D levels (27.8 ng/mL vs. 24.7 ng/mL, *p* = 0.09), and no association with dopamine transporter binding, suggesting that VitD insufficiency is not a preclinical marker [213]. Shrestha et al. analyzed data from 12,762 participants over 17 years and found no association between mid-life serum vitamin D levels and PD risk (HR = 1.05 for 20–30 ng/mL and HR = 1.14 for ≥30 ng/mL compared to <20 ng/mL) [214].

Luthra et al. evaluated the National Institutes of Health Exploratory Trials in Parkinson’s Disease (NET-PD) and found that only 12% of participants took more than 400 IU/day of VitD supplementation, and 34% used multivitamins estimated to contain 400 IU/day. Despite this, no significant differences in PD progression over three years were observed across groups taking VitD, multivitamins, both, or no supplements [215].

Ding et al. showed that the andrographolide derivative ZAV-12 (10–20 µM) acts as a *VDR* antagonist, significantly increasing LC3-II/LC3-I ratio and autophagy flux without affecting mTOR signaling, while also reducing A53T αSyn aggregation and lactate dehydrogenase release in SH-SY5Y cells, suggesting neuroprotective effects via SREBP2 activation [156].

Miyake et al. revealed that VitD intake showed no significant association with PD risk. The adjusted OR for the highest quartile of VitD intake (≥12.82 µg/day) compared to the lowest (<7.13 µg/day) was 0.82 (95% CI: 0.46–1.47; *p* for trend = 0.69), indicating no measurable protective or harmful effect [216].

The study by Wang et al. using bidirectional MR across three large PD cohorts does provide strong evidence that genetically predicted VitD levels are not causally associated with PD risk. By leveraging 131 high-quality SNPs related to VitD and excluding confounders like C-reactive protein and lipid levels, the authors found no significant associations using multiple MR methods, including IVW, MR-Egger, and weighted median approaches [15].

Larsson et al. suggests no causal link between VitD status and PD, as MR analysis using four SNPs associated with 25(OH)D3 showed no significant association with PD risk (OR per 10% lower 25OHD = 0.98; 95% CI: 0.93–1.04; *p* = 0.56). Although one SNP (rs6013897 in CYP24A1) showed a modest association with PD (OR = 1.09; *p* = 0.008), the overall genetic instrument did not support a causal role, indicating that previously observed lower VitD levels in PD patients may reflect reverse causation or confounding [217].

Anderson et al. showed that higher intake of VitD-rich foods was associated with an increased risk of PD, although this relationship appeared to be confounded by concurrent animal fat consumption [218]. Supporting this, Chen et al. found a significant association between dairy consumption—including dairy-derived VitD—and increased PD risk in men, but not in women, suggesting a gender-specific dietary influence [219]. A key limitation of both studies is the potential for dietary misclassification, particularly due to reliance on food frequency questionnaires that may not accurately capture portion sizes or long-term intake patterns.

Despite 60.5% of participants reporting pain, statistical analysis revealed no significant association between VitD levels and pain presence, suggesting that pain in PD may be influenced by factors other than VitD status [220].

Despite prior hypotheses and experimental evidence suggesting a neuroprotective role for VitD, several clinical studies have failed to demonstrate a consistent association between VitD levels and PD risk, severity, or symptomatology. One important consideration is the metabolic link between VitD and cholesterol. VitD synthesis in the skin relies on 7-dehydrocholesterol, a cholesterol precursor, and both pathways share biosynthetic mechanisms [221]. Consequently, many VitD-rich foods are also high in fat or derived from dairy products, which introduces a potential confounding factor. Notably, multiple epidemiological studies have reported a modest but significant association between high dairy consumption and increased PD risk, particularly in men [222].

Another limitation in many of the studies is the lack of randomization in sample selection. This introduces selection bias and may obscure true associations. Moreover, critical lifestyle factors such as physical activity and sun exposure—both of which influence VitD metabolism—were often not accounted for. Sunlight exposure, in particular, has been shown to correlate strongly with reduced PD risk, independent of VitD supplementation [223]. Taken together, these findings highlight the complexity of studying the role of VitD in neurodegeneration. Confounding dietary sources, metabolic interdependencies, and unmeasured lifestyle variables all contribute to the challenge of understanding the true effect of VitD on PD.

## 5. Future Studies

Despite the promising molecular and preclinical evidence supporting VitD’s role in neuroprotection and neural stem cell regulation, several limitations hinder its translation into clinical practice. A major concern is the lack of robust clinical validation; many findings are based on animal models or in vitro systems, and few have been confirmed through large-scale, randomized human trials. Additionally, the functional consequences of numerous *VDR* mutations remain poorly characterized, especially in the context of neurological diseases, making it difficult to predict individual responses to VitD-based therapies [224]. Supplementation studies have also yielded inconsistent results—some report cognitive improvements, while others show no significant benefit—highlighting the need for standardized dosing protocols and better stratification of patient populations. Furthermore, the long-term safety and efficacy of high-dose VitD or its analogs in neurodegenerative conditions are still uncertain, raising concerns about potential adverse effects such as hypercalcemia.

Molecular docking revealed that calcipotriol and inecalcitol exhibited strong binding affinities with p38 MAPK, engaging critical residues such as Tyr36 and Leu168, which are essential for kinase activity. Additionally, most VitD analogues, including calcitriol, were predicted to be blood–brain barrier permeable and non-neurotoxic [172]. Moreover, VitD was found to interact with 24 PD-related targets, with ESR2 emerging as a key hub. Molecular docking revealed a strong binding affinity between VitD and ESR2 (−8.6 kcal/mol), and Mendelian randomization analysis showed a significant protective effect of ESR2 activation on PD risk (OR = 0.46, 95% CI: 0.28–0.76, *p* = 0.002) [225].

### Clinical Trials

There are nine clinical trials registered in ClinicalTrials.gov assessing the effects of VitD on various PD outcomes (Figure 10) (Appendix A–Clinical Trials). Primary endpoints included motor and non-motor symptoms, cardiac autonomic dysfunction, EEG changes, balance, bone density, and immune modulation. Several trials focused on motor function using UPDRS and TUG, while others explored inflammation, cognitive performance, and gut microbiome. Notably, NCT06697626 and NCT06539260 included comprehensive secondary outcomes such as inflammatory markers and cognitive scales.

VitD therapy varied across studies, ranging from 400 IU BID to 50,000 IU weekly, with durations spanning 8 to 52 weeks. Some trials combined VitD with calcium or functional drinks, while others used BMI-adjusted dosing. High-dose regimens (e.g., 50 K IU/week) were common in short-term interventions, whereas lower daily doses were used in longer trials.

An interesting point to consider is that the true effect of VitD on neurodegeneration is extremely difficult to isolate and assess. Many patients included in the studies cited above were taking multivitamins containing VitD, and even those who are not currently supplementing may still be consuming VitD through fortified foods. As a result, determining the independent impact of VitD on neurodegenerative processes is inherently challenging.

Notably, the most recent guidelines from the Endocrine Society recommend VitD supplementation only for individuals aged 75 and older, suggesting that younger populations may not benefit from routine supplementation. This further complicates the design of studies aiming to evaluate VitD’s role in neuroprotection, as such research would likely require strict dietary control or even complete replacement of standard food sources to eliminate confounding variables.

As discussed earlier, numerous animal studies have demonstrated promising effects of VitD, including enhanced survival of dopaminergic neurons and slowed progression of neurodegeneration. However, these benefits have not been consistently replicated in clinical trials. This discrepancy may be due to the multifactorial nature of VitD’s effects, which are influenced not only by supplementation but also by lifestyle factors such as physical activity and sun exposure.

Therefore, indirect measures of VitD status may not fully capture its biological impact. While some studies have attempted to use sun exposure and UV-B radiation as proxies to better understand VitD’s role in neurodegenerative diseases, these approaches are also prone to variability and may yield unreliable results.

## 6. Conclusions

The current comprehensive review highlights a significant association between VitD deficiency and PD, with consistent evidence of lower serum levels and increased odds of insufficiency and deficiency among PD patients. VitD supplementation shows modest benefits in motor function, particularly balance and gait. Genetic analyses reveal that the FokI polymorphism in the *VDR* gene is most consistently linked to PD susceptibility, while other SNPs show variable or non-significant associations. Environmental factors such as outdoor exposure and dietary intake further influence PD risk, underscoring the multifactorial nature of VitD’s role in PD. Although experimental models support VitD’s neuroprotective mechanisms, clinical translation remains limited. Future studies should focus on longitudinal designs, standardized supplementation protocols, and mechanistic exploration to clarify VitD’s diagnostic, prognostic, and therapeutic potential in PD.

## Figures and Tables

**Figure 1 neurosci-06-00130-f001:**
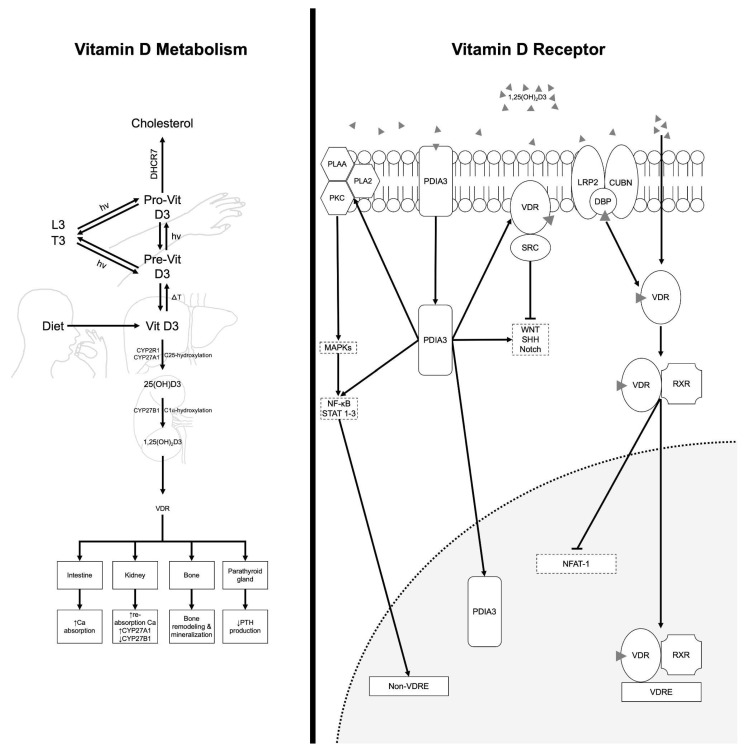
Vitamin D metabolism and receptor (VDR). Abbreviations: Ca, calcium; CUBN, cubilin; DBP, vitamin D binding protein; DHCR7, 7-dehydrocholesterol reductase; hv, light energy; L3, lumisterol 3; LRP2, low-density lipoprotein-related protein 2; MAPK, mitogen-activated protein kinase; NFAT1, nuclear factor of activated T cells; NF-κB, nuclear factor kappa-light-chain-enhancer of activated B cells; PDIA3 (ERp57), protein disulfide isomerase family A member 3; PKC, protein kinase C; PLAA, phospholipase A2 activating protein; PLA2, phospholipase A2; RXR, retinoid X receptor; SHH, sonic hedgehog; SRC, steroid receptor coactivator; STAT 1-3, signal transducers and activators of transcription 1 and 3; T3, tachysterol 3; VDR, vitamin D receptor; VDRE, vitamin D response element; WNT, wingless-type MMTV integration site family; △T, temperature change; ↑, increase/upregulate; ↓, decrease/downregulate.

**Figure 2 neurosci-06-00130-f002:**
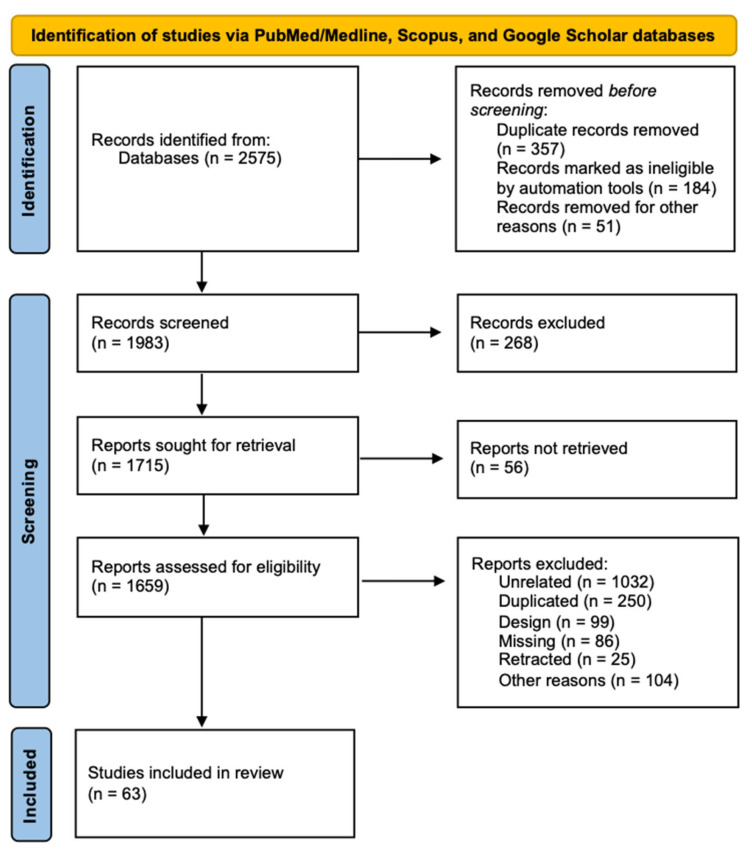
PRISMA flowchart for the identification of included studies.

**Figure 3 neurosci-06-00130-f003:**
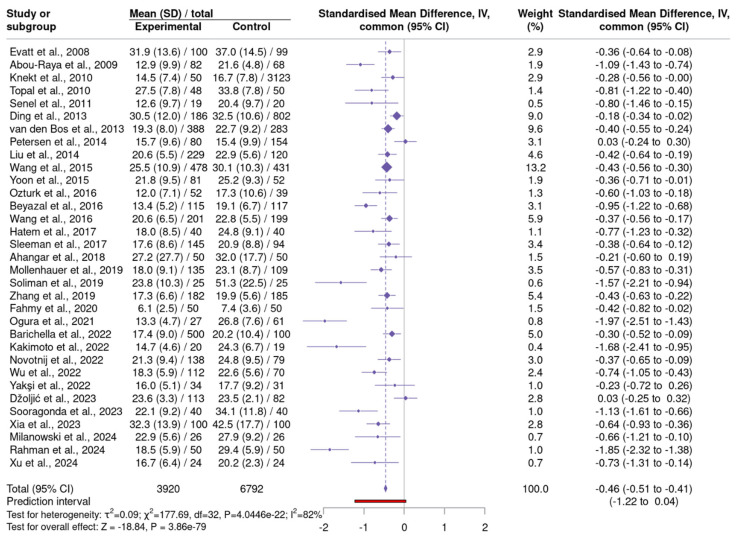
Forest plots of the vitamin D levels in patients with Parkinson’s disease versus controls.

**Figure 4 neurosci-06-00130-f004:**
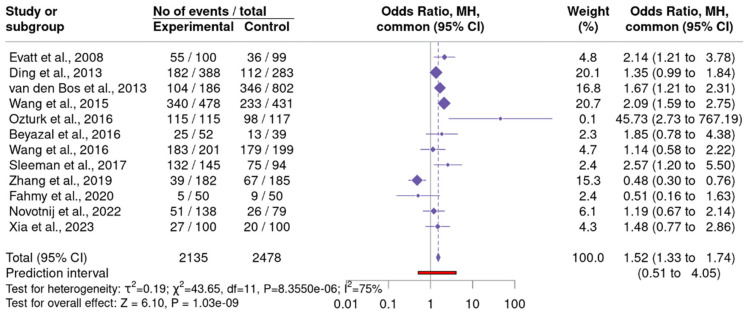
Forest plots of the vitamin D insufficiency in patients with Parkinson’s disease versus controls.

**Figure 5 neurosci-06-00130-f005:**
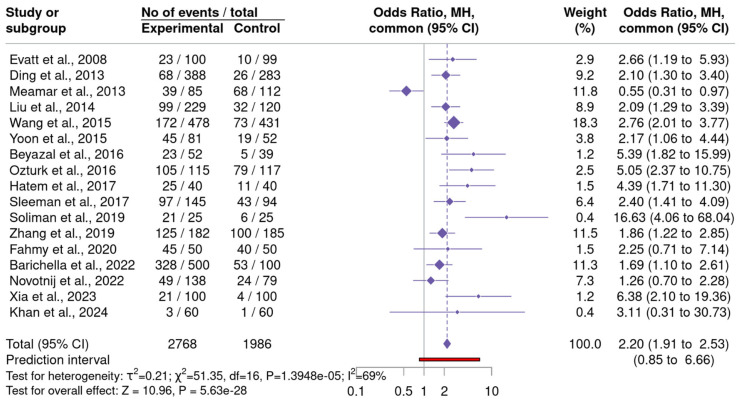
Forest plots of the vitamin D deficiency in patients with Parkinson’s disease versus controls.

**Figure 6 neurosci-06-00130-f006:**
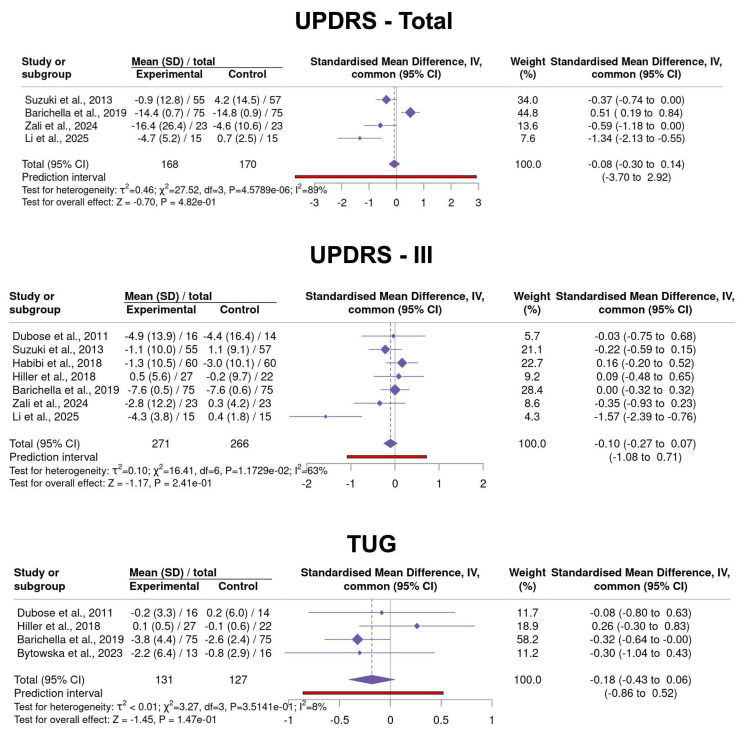
Forest plot of the effect of vitamin D on motor symptoms in patients with Parkinson’s disease compared to healthy controls. TUG, Timed Up and Go test; UPDRS, Unified Parkinson Disease Rating Scale; UPDRS—III, Part III (Motor) of UPDRS.

**Figure 7 neurosci-06-00130-f007:**
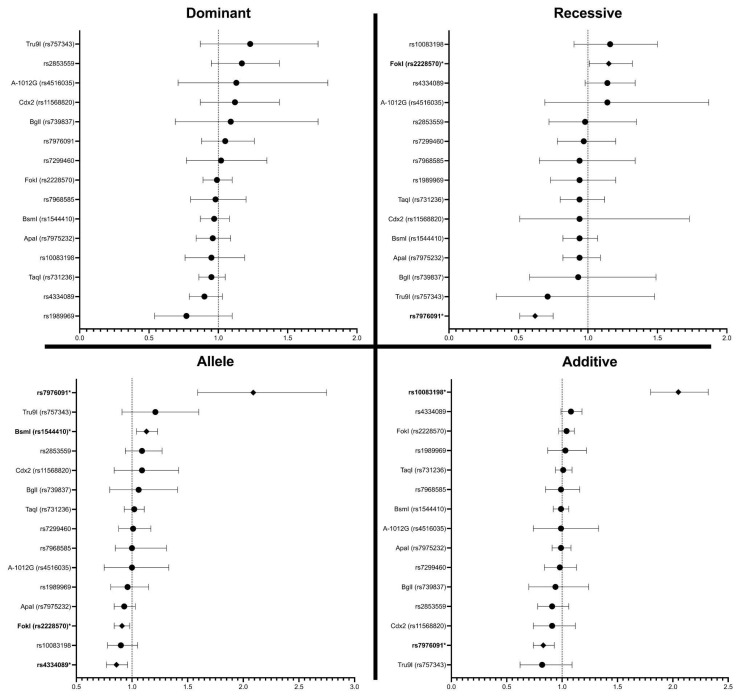
Forest plots showing odds ratios and 95% confidence intervals for vitamin d-related SNPs in Parkinson’s disease compared with healthy controls under allelic, dominant, recessive, and additive genetic models. Bolded results indicate statistically significant associations with *p* < 0.05.

**Figure 8 neurosci-06-00130-f008:**
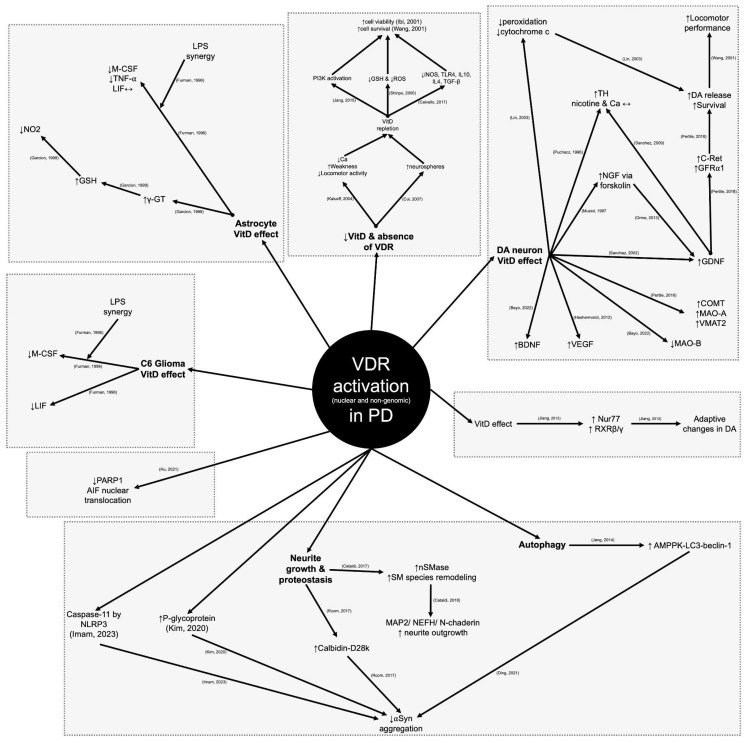
Vitamin D in Parkinson’s disease. Abbreviations: AIF, apoptosis-inducing factor; AMPK, adenosine monophosphate-activated protein kinase; BDNF, brain-derived neurotrophic factor; Ca, calcium; Calbindin-D28k, calcium-binding protein that buffers intracellular Ca^2+^; DA, dopamine; GDNF, glial cell derived neurotrophic factor; GFR⍺1, GDNF family receptor alpha-1; GSH, glutathione; IL4, interleukin 4; IL10, interleukin 10; iNOS, inducible nitric oxide synthase; LC3, microtubule-associated protein 1A/1B-light chain 3; LIF, leukemia inhibitory factor; LPS, lipopolysaccharide; MAO, monoamine oxidase; MAP2, microtubule-associated protein 2; M-CSF, macrophage colony-stimulating factor; NEFH, neurofilament heavy chain; NGF, nerve growth factor; NLRP3, nod-like receptor family, pyrin domain-containing 3; NO2, nitrogen dioxide; nSMase, neutral sphingomyelinase; Nur77, nuclear receptor 77; PARP1, poly(ADP-ribose) polymerase 1; PD, Parkinson’s disease; PI3K, phosphoinositide 3-kinase; ROS, reactive oxygen species; RXRβ/γ, retinoid X receptor isoforms β and γ; SM, sphingomyelin; TGF-β, transforming growth factor-beta; TH, tyrosine hydroxylase; TLR4, Toll-like receptor 4; TNF-α, tumor necrosis factor alpha; VDR, vitamin D receptor; VEGF, vascular endothelial growth factor; VitD, vitamin D; VMAT2, vesicular monoamine transporter 2; αSyn, α-synuclein; γ-GT, gamma-glutamyltransferase; ↑, increase; ↓, decrease; ↔, no effect, no dependent [7,8,120,121,122,123,124,125,126,127,128,129,130,131,132,133,134,135,136].

**Figure 9 neurosci-06-00130-f009:**
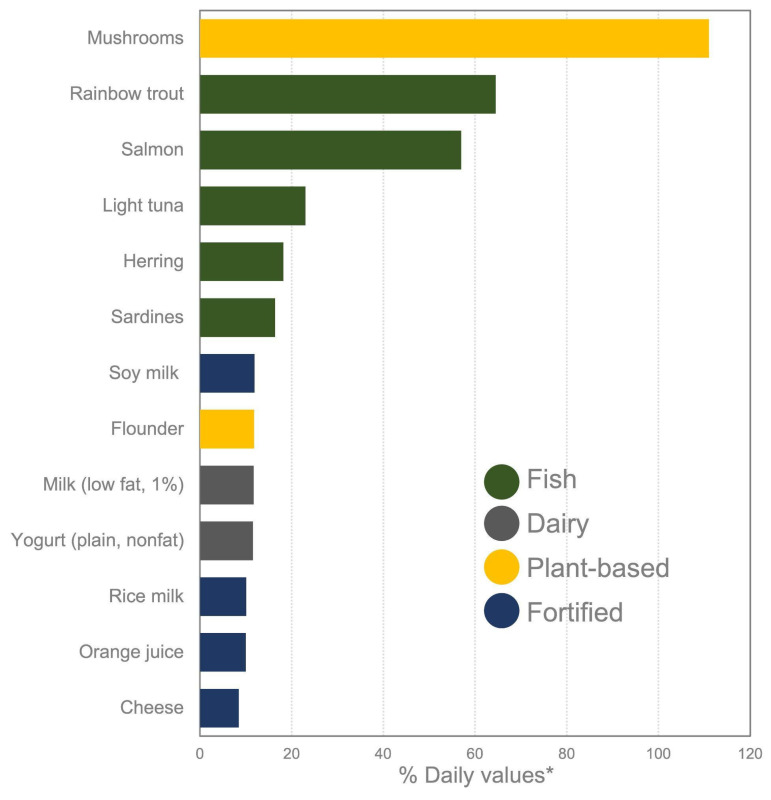
VitD content across selected food sources. * Recommended dietary allowance (1000 IU per day).

**Figure 10 neurosci-06-00130-f010:**
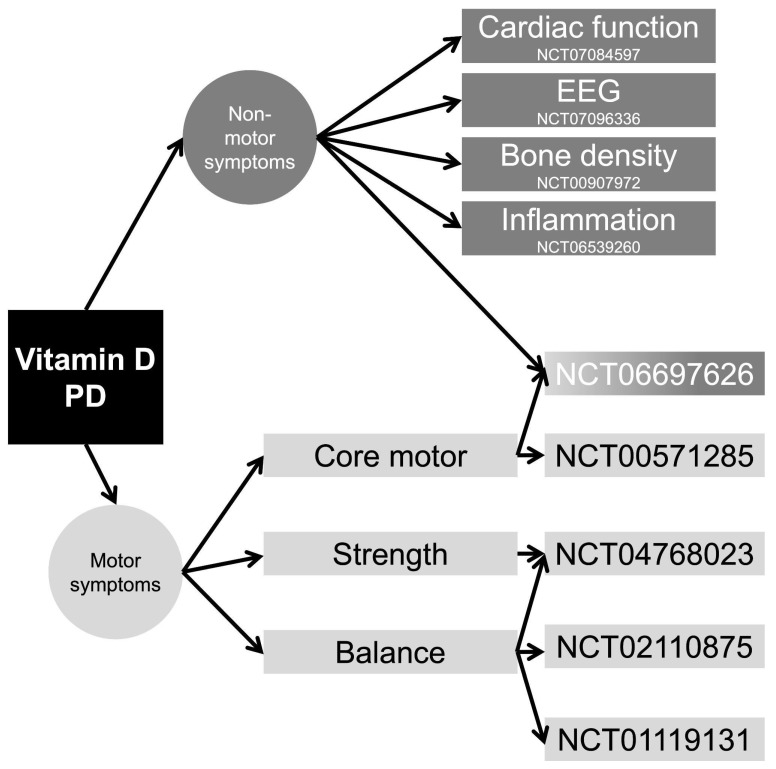
Clinical trials assessing the effect of vitamin D on Parkinson’s disease (PD). EEG, electroencephalogram.

**Table 1 neurosci-06-00130-t001:** Summary of associations between VitD and PD risk, symptoms, and genetic factors.

Category	Feature	Summary	References
PD risk	VitD serum	Sufficient VitD levels are associated with PD protection (HR 0.83)	Veronese et al. (2024) [91]
	*VDR*	Almost all *VDR* SNPs did not have correlation with PD, and had higher HWE	Read Figure 7
	PD vs. MSA	VitD levels associated with Klotho and homocysteine are able to differentiate PD than MSA (AUC = 0.81, SN 71%, SP 78%)	Guo et al. (2017) [93]
	Genetic	VitD deficiency was correlated to non-genetic cases of PD (β = −0.61)	Thaler et al. (2021) [94]
Motor symptoms	Severity	VitD deficiency more commonly found in individuals with severe PD	Ahangar et al. (2018) [45]
	Age of onset	VitD was negatively correlated with PD age of onset (r = −0.29)	Fahmy et al. (2020) [49]
	Freezing	Freezing is more common in VitD deficiency	Moghaddasi et al. (2013) [95]
	Balance and falls	Low VitD associated with falls; high-dose supplements may help younger patients.	Peterson et al. (2013) [96]
Non-motor symptoms	Cognition	Higher VitD levels are associated with better verbal fluency (β = 0.26) and memory (β = 0.19)	Peterson et al. (2013) [97]
	Fatigue	Fatigue severity was correlated with lower VitD levels (r = −0.85)	Canlı et al. (2024) [98]
	Mood	Worsening depression and anxiety in PD patients with VitD deficiency	Zhang et al. (2019) [48]
	OH	Patients with VitD deficiency more commonly have OH (OR 39.56)	Jang et al. (2015) [99]
	GI autonomic dysfunction	VitD deficiency was correlated with gastroparesis (r = −0.34)	Kwon et al. (2016) [100]
	Olfactory	Lower VitD correlated with poorer smell identification (β = 0.38)	Kim et al. (2018) [101]
Scales	UPDRS	VitD serum negatively correlated to UPDRS scores (r = −0.34)	Meamar et al. (2015) [102]
	UPDRS-II	VitD levels were correlated to worse baseline activities of daily living (β_0_ − 0.75)	Lawton et al. (2020) [103]
	UPDRS-III	VitD was correlated with worsening motor symptoms (β − 2.79)	Barichella et al. (2020) [51]
fMRI	fMRI	More commonly low-frequency fluctuation in PD patients with VitD deficiency	Lv et al. (2021) [104]

Abbreviations: HR, hazard ratio; OH, orthostatic hypotension; OR, odds ratio; PD, Parkinson’s disease; r, spearman’s correlation coefficient; SN, sensitivity; SNP, single nucleotide polymorphism; SP, specificity; UPDRS, Unified Parkinson’s Disease Rating Scale; VDR, vitamin D receptor; VitD, vitamin D; HWE, Hardy–Weinberg equilibrium; β, standardized regression coefficient from a multiple linear regression analysis; β_0_, intercept.

**Table 2 neurosci-06-00130-t002:** Genes roles in dopaminergic neuron function and response to VitD.

Gene	Chromosomal Location	Role	Response	Reference
*C-Ret*	10q11.2	Supports neuronal survival and antioxidant defense	↑	Pertile et al. (2018) [133]
*GDNF*	5p13	Promotes dopaminergic neuron health and neurotransmission	↑	Pertile et al. (2018) [133]
*Nurr1*	2q22–23	Essential for dopamine neuron development and maintenance	↑	Cui et al. (2010) [144]
*p57kip2*	11p15.5	Involved in differentiation and maturation of dopamine neurons	↑	Cui et al. (2010) [144]
*SLC30A10*	2q32.3	Regulates metal ion balance (Ca^2+^, Zn^2+^, Fe^2+^, Mn^2+^)	↑	Claro da Silva et al. (2016) [163]
*SLC39A2*	14q11.2	Participates in metal ion homeostasis	↓	Claro da Silva et al. (2016) [163]
*TH*	11p15	Key enzyme in dopamine synthesis	↑	Jiang et al. (2014) [164]
*BST-1 (CD157)*	4p15	Associated with vulnerability of dopaminergic neurons	↑	Dygaĭ et al. (1998) [165]
*GAK*	4p16	Involved in neuronal development	↑	Kesavapany et al. (2004) [166]
*STBD1*	4q21.1	Brain function not yet established	↔	Carpenter et al. (1989) [167]
*HLA-DRA*	6p21.3	Brain role remains unclear	↓	Carpenter et al. (1989) [167]
*SFXN2*	10q24.32	Expressed in neural ectodermal tissues	↔	Carpenter et al. (1989) [167]
*LRRK2*	12q12	Plays a role in neurogenesis and neurite extension	↔	Kattar et al. (2023) [168]

Abbreviations: ↑, increase/upregulate; ↓, decrease/downregulate; ↔, no effect, no dependent, unclear.

**Table 3 neurosci-06-00130-t003:** Vitamin D supplementation summary according to the last guideline by the Endocrine Society [208].

Population Group	Status	Recommended Dose	Goal Serum Level
General Adults (Adults to 74 yrs)	Routine intake	50–70 yrs: 600 IU/day; against routine supplementation	Maintain ≥ 20 ng/mL
		70–74 yrs: 800 IU/day	Maintain ≥ 20 ng/mL
	Deficient (<29 ng/mL)	1000–2000 IU/day or 50,000 IU/week × 6–8 wks	30–40 ng/mL
	Obese and Deficient	80,000–100,000 IU/week × 6 wks, then 2000–4000 IU/day	≥30 ng/mL
	Diabetes and pre-diabetes	3500 IU/day	≥30 ng/mL
Older Adults (≥75 yrs)	Routine intake	~900 IU/day; suggested empiric supplementation ^a^	Maintain ≥ 20 ng/mL
	Deficient (<20 ng/mL)	2000 IU/day or 50,000 IU/week × 8 wks	30–40 ng/mL
	Obese and Deficient	80,000–100,000 IU/week × 6 wks, then 2000–4000 IU/day	≥30 ng/mL

Considerations: (i) Routine testing for vitamin D is not advised in any group. (ii) Vitamin D3 (cholecalciferol) is generally preferred over D2. (iii) Consider co-supplementation with calcium (e.g., 1000 mg/day) in older adults for bone health. (iv) Doses of 50,000–100,000 IU/week should only be given in clear deficiency, and should be recommended daily doses as soon as able to obtain normal vitamin D levels. Daily doses should always be preferred. ^a^ small but significant overall reduction in mortality.

## Data Availability

The original contributions presented in this study are included in the article/Appendix A. Further inquiries can be directed to the corresponding author(s).

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
