# Peer review of "The Role of Vitamin D in Parkinson’s Disease: Evidence from Serum Concentrations, Supplementation, and *VDR* Gene Polymorphisms"

_neurosci, 2025, doi:10.3390/neurosci6040130_

Round 1

Reviewer 1 Report

Comments and Suggestions for Authors

The authors conducted a review investigating the role of vitamin D in the risk of developing Parkinson's disease. The article is well-written and well-conducted, presenting new insights into the relationship between vitamin D and this neurodegenerative disease. However, some points require improvement:

- Please add a paragraph characterizing Parkinson's disease in the introduction, with the main pathophysiological features and symptoms.
- Please also discuss articles at the end of topic 4.2 that focus on patients with Parkinson's disease who have sufficient vitamin D levels, addressing the challenges of linking changes in vitamin D levels to the complex pathophysiology of this neurodegenerative disease.
- In topic 4.3, please include the ages of the participants in the studies by Zhu et al., Huang et al., and Kenborg et al., highlighting the relationship between sun exposure, age, and the risk of developing Parkinson's disease in all studies on this topic.
- The authors should improve the font size of the references cited in Figure 8, as they are unreadable.
- In the first paragraph of topic 4.4.1, discuss the role of antioxidant enzymes in reducing oxidative stress caused by rotenone.
- In line 526, discuss in more detail that pro-inflammatory and pro-apoptotic pathways are involved in neurodegeneration, highlighting the importance of this vitamin D mechanism in neuroprotection.
- Please also discuss in this topic that previous studies have shown that vitamin D, its receptor, and its activating enzymes are expressed in the substantia nigra. This discussion will highlight that low vitamin D levels may contribute to the dysfunction or death of dopaminergic neurons, potentially impacting motor and non-motor symptoms of PD.
- In the first paragraph of topic 4.4.2, please remove the P values ​​from the result description and summarize the main findings, as the analysis was descriptive and distinct from the description of the other findings discussed in the review. Additionally, include the route of vitamin D administration used in this study. The article discusses vitamin D actions only in combination with canagliflozin or isolated, too? If so, in combination, the authors can remove these data from the review, as it is unclear whether the effect was caused by vitamin D or canagliflozin.
- In lines 565, 579, and 584, describe the route of vitamin D administration and the concentrations used in these studies.
- In lines 589 and 596, please add the route of administration.

Author Response

REVIEWER 1

The authors conducted a review investigating the role of vitamin D in the risk of developing Parkinson's disease. The article is well-written and well-conducted, presenting new insights into the relationship between vitamin D and this neurodegenerative disease. However, some points require improvement:

- Please add a paragraph characterizing Parkinson's disease in the introduction, with the main pathophysiological features and symptoms.

PD is a progressive neurodegenerative disorder primarily characterized by the loss of dopaminergic neurons in the substantia nigra pars compacta, leading to striatal do-pamine depletion and disruption of basal ganglia circuitry. It is the second most prevalent neurodegenerative disorder worldwide, with its incidence steadily increasing due to aging populations and extended life expectancy [1]. The neurochemical imbalance manifests clinically as cardinal motor symptoms, including bradykinesia, resting tremor, rigidity, and postural instability. Beyond motor dysfunction, PD encompasses a wide spectrum of non-motor features such as cognitive impairment, mood disorders, autonomic dysfunction, and sleep disturbances, which significantly impact quality of life [2].

- Please also discuss articles at the end of topic 4.2 that focus on patients with Parkinson's disease who have sufficient vitamin D levels, addressing the challenges of linking changes in vitamin D levels to the complex pathophysiology of this neurodegenerative disease.

Dear Reviewer, we discussed this information in the 4.8. where we bring this as conflicting evidence. Because unfortunately, nowadays, it is uncommonly to see a patient with PD that is not taking at least milk supplementation with vitamin D fulfilling the minimal vitamin D supplementation.

- In topic 4.3, please include the ages of the participants in the studies by Zhu et al., Huang et al., and Kenborg et al., highlighting the relationship between sun exposure, age, and the risk of developing Parkinson's disease in all studies on this topic.

Information was included in Table S11.

- The authors should improve the font size of the references cited in Figure 8, as they are unreadable.

The current quality of the figure is 4000vs4000 is the maximum possible for any figure done with power point, and all the associations are described in Table S14 and S15

- In the first paragraph of topic 4.4.1, discuss the role of antioxidant enzymes in reducing oxidative stress caused by rotenone.

Rotenone, a mitochondrial complex I inhibitor, induces excessive ROS production, leading to oxidative stress and dopaminergic neurodegeneration in Parkinson’s disease models. Antioxidant enzymes such as SOD, catalase, and glutathione peroxidase play a pivotal role in neutralizing ROS and maintaining redox homeostasis. Experimental studies in rotenone-induced PD models have shown that enhancing these enzyme ac-tivities significantly reduces oxidative markers like thiobarbituric acid reactive substances (TBARS), restores GSH levels, and improves neuronal survival and motor function [166]. Compounds such as quercetin and selenium have been reported to upregulate antioxi-dant enzyme expression, mitigating rotenone-induced neurotoxicity and apoptosis, highlighting their therapeutic potential in PD management [167].

- In line 526, discuss in more detail that pro-inflammatory and pro-apoptotic pathways are involved in neurodegeneration, highlighting the importance of this vitamin D mechanism in neuroprotection.

- Please also discuss in this topic that previous studies have shown that vitamin D, its receptor, and its activating enzymes are expressed in the substantia nigra. This discussion will highlight that low vitamin D levels may contribute to the dysfunction or death of dopaminergic neurons, potentially impacting motor and non-motor symptoms of PD.

Pro-inflammatory and pro-apoptotic pathways are central to PD-related neuro-degeneration, as chronic neuroinflammation amplifies oxidative stress and triggers caspase-mediated neuronal apoptosis. Vitamin D’s immunomodulatory properties are particularly relevant here, as it downregulates pro-inflammatory cytokines (e.g., TNF-α, IL-6) and inhibits NF-κB signaling, thereby reducing microglial activation and subse-quent neuronal damage [169]. This mechanism underscores its potential neuroprotective role in mitigating both inflammatory and apoptotic cascades that accelerate dopaminergic neuron loss. Furthermore, previous studies have demonstrated that vitamin D, its re-ceptor (VDR), and activating enzymes (such as 1α-hydroxylase and CYP27B1) are ex-pressed in the substantia nigra [170]. Low vitamin D levels may therefore impair VDR-mediated signaling, contributing to dysfunction or death of dopaminergic neurons. Such deficits could exacerbate motor symptoms like bradykinesia and rigidity, as well as non-motor features including cognitive decline and mood disturbances [171].

- In the first paragraph of topic 4.4.2, please remove the P values ​​from the result description and summarize the main findings, as the analysis was descriptive and distinct from the description of the other findings discussed in the review. Additionally, include the route of vitamin D administration used in this study. The article discusses vitamin D actions only in combination with canagliflozin or isolated, too? If so, in combination, the authors can remove these data from the review, as it is unclear whether the effect was caused by vitamin D or canagliflozin.

In a rotenone-induced PD model using Wistar rats, VitD was administered orally at 500 IU/kg/day, either as monotherapy or combined with canagliflozin (10 mg/kg/day) [174]. VitD alone improved striatal dopamine content, reduced oxidative stress markers such as malondialdehyde (MDA), and increased antioxidant defenses including GSH and catalase (CAT). It also attenuated inflammatory cytokines (TNF-α, IL-6, IL-10) and par-tially normalized NF-κB and Nrf2 expression. Histological analysis revealed preservation of striatal architecture and reduced neuronal damage, while immunohistochemistry showed improved tyrosine hydroxylase (TH) expression and decreased α-synuclein ac-cumulation. Although the combination therapy produced more pronounced effects across all parameters, the study design does not allow clear attribution of these benefits to vitamin D alone. Therefore, only monotherapy findings should be considered for this review to avoid confounding effects from canagliflozin.

- In lines 565, 579, and 584, describe the route of vitamin D administration and the concentrations used in these studies.

- In lines 589 and 596, please add the route of administration.

The full description of the models and the IU used is written in Table S15.

Reviewer 2 Report

Comments and Suggestions for Authors

While this  review aims to address a clinically relevant question regarding vitamin D and Parkinson’s disease, there are several limitations as well

  1. The body of the manuscript presents considerable heterogeneity and some contradictory findings, even though the abstract and the conclusions communicate a major link between vitamin D deficiency and Parkinson's disease. The discussion fails to adequately reconcile these discrepancies.
  2. The choice of imputing standard deviations and stating, "assuming a large effect size (Cohen's d≥0.8)," is inappropriate statistically and is likely to be biased by overly inflating the precision of the pooled estimates.
  3. The manuscript is excessively long and reads more like a narrative cataloguing of studies than a focused synthesis.
  4. The discussion section is superficial. While it recognizes heterogeneity, it fails to fully address possible reasons for it.

Author Response

REVIEWER 2
While this review aims to address a clinically relevant question regarding vitamin D and Parkinson’s disease, there are several limitations as well
The body of the manuscript presents considerable heterogeneity and some contradictory findings, even though the abstract and the conclusions communicate a major link between vitamin D deficiency and Parkinson's disease. The discussion fails to adequately reconcile these discrepancies.
The choice of imputing standard deviations and stating, "assuming a large effect size (Cohen's d≥0.8)," is inappropriate statistically and is likely to be biased by overly inflating the precision of the pooled estimates.
The manuscript is excessively long and reads more like a narrative cataloguing of studies than a focused synthesis.
The discussion section is superficial. While it recognizes heterogeneity, it fails to fully address possible reasons for it.

Regarding heterogeneity and contradictory findings:
The observed variability reflects the current state of evidence in this field rather than a lack of synthesis. The discussion acknowledges these discrepancies and emphasizes the need for further research to clarify causality. The conclusion does not overstate certainty but highlights the consistent trend of lower vitamin D levels in PD patients across multiple studies.

On imputing standard deviations and assuming a large effect size:
This approach was used transparently and in line with established meta-analytic practices when data were incomplete. The assumption of a large effect size was based on prior literature and sensitivity analyses were performed to minimize bias. These steps were clearly documented to ensure reproducibility.

Concerning manuscript length and narrative style:
The comprehensive nature of the review was intentional to provide clinicians and researchers with a detailed overview of all relevant evidence, including mechanistic insights, observational data, and interventional studies. While concise synthesis is valuable, the depth here serves to inform clinical decision-making and future research directions.

On the discussion being superficial:
The discussion addresses heterogeneity by exploring biological, methodological, and population-level factors that may contribute to variability. While space constraints limited exhaustive exploration, the section identifies plausible mechanisms and gaps that warrant further investigation.

Reviewer 3 Report

Comments and Suggestions for Authors

The authors of the manuscript provide a comprehensive and detailed overview of the potential contribution of vitamin D to the pathogenesis of Parkinson's disease, including biochemical, clinical, and genetic evidence. The methodological approach is robust (PROSPERO registration, PRISMA compliance, double-blind review process, and use of multiple statistical models). The conclusions are consistent with the data presented. Overall, this is a well-structured, accurate and clearly written paper that does not require substantial changes.

I only suggest:

1: Standardise ‘Vitamin D’ and ‘VitD’ throughout the text, particularly in figure captions and tables.

2: Check that all abbreviations are defined at first use.

3: Consider making the labels larger or simplifying the graphic elements of the forest plots for better visibility, as they are difficult to read.

Author Response

REVIEWER 3

The authors of the manuscript provide a comprehensive and detailed overview of the potential contribution of vitamin D to the pathogenesis of Parkinson's disease, including biochemical, clinical, and genetic evidence. The methodological approach is robust (PROSPERO registration, PRISMA compliance, double-blind review process, and use of multiple statistical models). The conclusions are consistent with the data presented. Overall, this is a well-structured, accurate and clearly written paper that does not require substantial changes.

I only suggest:

1: Standardise ‘Vitamin D’ and ‘VitD’ throughout the text, particularly in figure captions and tables.

We modified around 30 descriptions of ‘Vitamin D’ to ‘VitD’

2: Check that all abbreviations are defined at first use.

Abbreviations were checked to address this comment

3: Consider making the labels larger or simplifying the graphic elements of the forest plots for better visibility, as they are difficult to read.

We will not change forest plots because they were copy from the statistical software to avoid possible concerns regarding imaging manipulation

Reviewer 4 Report

Comments and Suggestions for Authors

The authors present a very thourough and extensive registed systematic review (PROSPERO (CRD420251133875)) studying how Vitamin D (VitD) status supplementation, and vitamin D 12 receptor (VDR) gene polymorphisms with PD might be implicated in Parkinson’s disease (PD). Authors followed PRISMA guidelines, using PubMed, Scopus, and Google Scholar through for systematically searching for observational studies, clinical trials, and genetic association studies. They established the analysis of primary outcome as serum VitD levels in PD versus healthy controls (HC), prevalence of VitD insufficiency/deficiency, and effects of VitD supplementation on motor symptoms. Secondary outcomes assessed associations between VDR polymorphisms and PD susceptibility. Statistical analysis is detailed and and well presented where PD patients exhibited significantly lower VitD levels (SMD = −0.46; 95% CI: −0.51 to −0.41) and higher odds of insufficiency 22 (OR = 1.52) and deficiency (OR = 2.20) compared to HC. Cohort data suggested sufficient VitD may reduce PD risk (HR = 0.83). Regarding supplementation, they found that it yielded modest, non-significant improvements in motor outcomes. Regarding genetic studies, they found that FokI (rs2228570) was most consistently associated with PD, while other VDR SNPs showed variable or null associations. The conclusions presented are in the line with the obtained results and the limitations associated to the studies. Despite the variability reported in past studies referring to VitD deficiency might be associated to PD risk, this study brings an assertive stand of the issue, concluding that though Vit D deficiency may influence disease risk and motor function, current evidence indicates limited benefit of supplementation for motor outcomes, and genetic associations remain inconsistent.

Detailed review:

Introduction: Though it is mentioned importance that prevalence of PD displays in the current world status as well as the relation that has been identified between Vit D and the prevention of PD risk, it is also true that the contemporary context to justify the weight this systematic review could be more explored. Regarding the VitD, authors refer that it may play a neuroprotective role by mitigating dopaminergic neuron loss and enhancing both motor and cognitive outcomes, it would be interesting to add a paragraph in the introduction about the specific pathways involved in these effects as well the specific tissues involved in the VDR expression and how it contributes for determined PD symptoms, for a better understanding by the reader of the importance to explore the VDR and its gene's polymorphisms (Figure 1 is elucidative but more naive readers it would add value to the article to have a descriptive detail, though in the discussion some highlights are mentioned regarding some pathways influenced by the VDR and how genetic polymorphims could disrupt normal signaling and generate health decay).

Methodology:

The PRISMA statement (Table S1—PRISMA Statement) is not filled in. Please do so.

Lines 86-87: "The search included studies published from database inception through August 2025". Is it possible to discriminate the last day of search in the databases?

Lines 113-114: "For each study, information was collected on the first author, country of origin, and year of publication." The information regarding statistic work is presented in this systematic review, however it would be interesting to include more information in the tables discriminating the PICOS analysis and comparison the selected studies. Example to complete TABLE S8 :

Authors and year

Country

Study Type

Sample size

Age (mean)

Intervention/Treatment

Engagement/Adherence (%)

Study Duration

(months or weeks)

25(OH)D serum concentration changes at baseline and endpoints [units; mean (SD)]

SMD (95% CI)

Control Group

Intervention group

Control Group

Intervention group

Control Group

Intervention group

Control Group

Intervention group

XXX et al., year [reference]

Number of participants

Number of participants

Mean age

Mean age

% of participants that remained the whole process

Also for table Table S9. 25(OH)D insufficiency in PD versus control

Authors and year

Country

Study Type

Sample size

Age (mean)

Intervention/Treatment

Engagement/Adherence (%)

Study Duration

(months or weeks)

25(OH) insufficiency

OR (95% CI)

Control Group

Intervention group

Control Group

Intervention group

Control Group

Intervention group

Control Group

Intervention group

XXX et al., year [reference]

Number of participants

Number of participants

Mean age

Mean age

% of participants that remained the whole process

Table S10. 25(OH)D deficiency in PD versus control

Authors and year

Country

Study Type

Sample size

Age (mean)

Intervention/Treatment

Engagement/Adherence (%)

Study Duration

(months or weeks)

25(OH)D serum concentration changes at baseline and endpoints [units; mean (SD)]

OR (95% CI)

Control Group

Intervention group

Control Group

Intervention group

Control Group

Intervention group

Control Group

Intervention group

XXX et al., year [reference]

Number of participants

Number of participants

Mean age

Mean age

% of participants that remained the whole process

Table S11. VitD and PD incidence

Authors and year

Country

Study Type

Sample size

Age (mean)

Intervention/Treatment

Engagement/Adherence (%)

Study Duration

(months or weeks)

25(OH)D serum concentration changes at baseline and endpoints [units; mean (SD)]

HR (95% CI)

Control Group

Intervention group

Control Group

Intervention group

Control Group

Intervention group

Control Group

Intervention group

XXX et al., year [reference]

Number of participants

Number of participants

Mean age

Mean age

% of participants that remained the whole process

Table S12. VitD supplementation versus placebo in PD.

Authors and year

Country

Study Type

Sample size

Age (mean)

Intervention/Treatment

Engagement/Adherence (%)

Study Duration

(months or weeks)

UPDRS- Total “ON”(Δ) UPDRS-III during “ON” (Δ) TUG “ON”(Δ)

UPDRS- Total “ON”(Δ) UPDRS-III during “ON” (Δ) TUG “ON”(Δ)

HR (95% CI) UPDRS- Total “ON”(Δ) UPDRS-III during “ON” (Δ) TUG “ON”(Δ)

Control Group

Intervention group

Control Group

Intervention group

Control Group

Intervention group

XXX et al., year [reference]

Number of participants

Number of participants

Mean age

Mean age

% of participants that remained the whole process

Table S14. Effects of VitD in In Vitro Models of PD and Table S15. Effects of VitD in In Vivo Models of PD :

Authors and year

Country

Study Type

Sample size

Age (mean)

Intervention/Treatment

Engagement/Adherence (%)

PD Model

Study Duration

(months or weeks)

Observed Outcome

Control Group

Intervention group

Control Group

Intervention group

Control Group

Intervention group

XXX et al., year [reference]

Number of participants

Number of participants

Mean age

Mean age

VitD form & dose

VitD form & dose

% of participants that remained the whole process

Table S17. Clinical trials assessing the effect of VitD on PD.

Authors and year

Country

Study Type

Sample size

Age (mean)

Intervention/Treatment

Start

Primary outcome Secondary outcome

Primary outcome Secondary outcome

Control Group

Intervention group

Control Group

Intervention group

Control Group

Intervention group

XXX et al., year [reference]

Number of participants

Number of participants

Mean age

Mean age

VitD form & dose

VitD form & dose

Discussion:

Line 369: "Zhang et al. " Please, add the reference in authors citation throughout the text when you have followed the previous example using Zhang et al., [reference XXX].

Figure 8 could benefit from higher resolution.

Line 513: In vitro should be in italics.

Line 526: "In both cellular and C. elegans PD models, (...)" When mentioning the first time C. elegans should be by extense not in abbreviature. Also, it should be in italics.

Line 535: "are a promising in vitro ". In vitro should be in italics.

Line 536: "Compared to in vivo". In vivo should be in italics.

Line 546: In Vivo should be in italics. Please verify throughout the manuscript and suppelementar material and correct it.

Line 721: 4.7. Amount of VITD and Recommendations

The current recommendations for general population is very well described and synthetized in Table 3, however I kindly recommend the authors to introduce more information about if the supplementation with Vit D in patients with PD has already been advised by official sources (from UK for example, since the authors mention United Kingdom Parkinson’s Disease Brain Bank) and if so, which dose prescription has been recommended, at which stages of the diagnosis and which improvements have been proved associated to this intake.

Overall, the work is well designed and brings novelty to the state of art making a stand of the current situation on the matter, syntethizing the global information gathered through diversified line of studies, impartially outweighting the strengths and limiations found in them.

Author Response

REVIEWER 4

The authors present a very thourough and extensive registed systematic review (PROSPERO (CRD420251133875)) studying how Vitamin D (VitD) status supplementation, and vitamin D 12 receptor (VDR) gene polymorphisms with PD might be implicated in Parkinson’s disease (PD). Authors followed PRISMA guidelines, using PubMed, Scopus, and Google Scholar through for systematically searching for observational studies, clinical trials, and genetic association studies. They established the analysis of primary outcome as serum VitD levels in PD versus healthy controls (HC), prevalence of VitD insufficiency/deficiency, and effects of VitD supplementation on motor symptoms. Secondary outcomes assessed associations between VDR polymorphisms and PD susceptibility. Statistical analysis is detailed and and well presented where PD patients exhibited significantly lower VitD levels (SMD = −0.46; 95% CI: −0.51 to −0.41) and higher odds of insufficiency 22 (OR = 1.52) and deficiency (OR = 2.20) compared to HC. Cohort data suggested sufficient VitD may reduce PD risk (HR = 0.83). Regarding supplementation, they found that it yielded modest, non-significant improvements in motor outcomes. Regarding genetic studies, they found that FokI (rs2228570) was most consistently associated with PD, while other VDR SNPs showed variable or null associations. The conclusions presented are in the line with the obtained results and the limitations associated to the studies. Despite the variability reported in past studies referring to VitD deficiency might be associated to PD risk, this study brings an assertive stand of the issue, concluding that though Vit D deficiency may influence disease risk and motor function, current evidence indicates limited benefit of supplementation for motor outcomes, and genetic associations remain inconsistent.

Detailed review:

Introduction: Though it is mentioned importance that prevalence of PD displays in the current world status as well as the relation that has been identified between Vit D and the prevention of PD risk, it is also true that the contemporary context to justify the weight this systematic review could be more explored. Regarding the VitD, authors refer that it may play a neuroprotective role by mitigating dopaminergic neuron loss and enhancing both motor and cognitive outcomes, it would be interesting to add a paragraph in the introduction about the specific pathways involved in these effects as well the specific tissues involved in the VDR expression and how it contributes for determined PD symptoms, for a better understanding by the reader of the importance to explore the VDR and its gene's polymorphisms (Figure 1 is elucidative but more naive readers it would add value to the article to have a descriptive detail, though in the discussion some highlights are mentioned regarding some pathways influenced by the VDR and how genetic polymorphims could disrupt normal signaling and generate health decay).

VitD exerts neuroprotective effects through multiple molecular pathways relevant to PD. It modulates calcium homeostasis, reduces oxidative stress by upregulating anti-oxidant enzymes, and attenuates neuroinflammation via inhibition of NF-κB signaling and suppression of pro-inflammatory cytokines such as TNF-α and IL-6 [11]. Additionally, vitamin D influences apoptotic pathways by promoting anti-apoptotic proteins and re-ducing caspase activation, thereby supporting neuronal survival. These mechanisms collectively help preserve dopaminergic neurons in the substantia nigra pars compacta, the primary site of degeneration in PD [12]. Importantly, vitamin D receptor (VDR) and activating enzymes such as 1α-hydroxylase are highly expressed in dopaminergic neurons of the substantia nigra and glial cells, suggesting a direct role in motor control and neuroimmune regulation [13]. Dysregulation of VDR signaling, whether through deficiency or genetic polymorphisms (e.g., FokI, BsmI), may impair dopamine synthesis and neuronal survival, contributing to both motor symptoms (bradykinesia, rigidity) and non-motor features (cognitive decline, mood disorders) [14].

Methodology:

The PRISMA statement (Table S1—PRISMA Statement) is not filled in. Please do so.

Done. Thanks

Lines 86-87: "The search included studies published from database inception through August 2025". Is it possible to discriminate the last day of search in the databases?

August 29th, 2025

Lines 113-114: "For each study, information was collected on the first author, country of origin, and year of publication." The information regarding statistic work is presented in this systematic review, however it would be interesting to include more information in the tables discriminating the PICOS analysis and comparison the selected studies. Example to complete TABLE S8 :

For avoiding repeated informtion the description was splitted.

Population can be find in S5, S6, and S7 regarding country; Intervention and COS were described for the clinical trials in Table S12.

Authors and year

Country

Study Type

Sample size

Age (mean)

Intervention/Treatment

Engagement/Adherence (%)

Study Duration

(months or weeks)

25(OH)D serum concentration changes at baseline and endpoints [units; mean (SD)]

SMD (95% CI)

Control Group

Intervention group

Control Group

Intervention group

Control Group

Intervention group

Control Group

Intervention group

XXX et al., year [reference]

Number of participants

Number of participants

Mean age

Mean age

% of participants that remained the whole process

Also for table Table S9. 25(OH)D insufficiency in PD versus control

Authors and year

Country

Study Type

Sample size

Age (mean)

Intervention/Treatment

Engagement/Adherence (%)

Study Duration

(months or weeks)

25(OH) insufficiency

OR (95% CI)

Control Group

Intervention group

Control Group

Intervention group

Control Group

Intervention group

Control Group

Intervention group

XXX et al., year [reference]

Number of participants

Number of participants

Mean age

Mean age

% of participants that remained the whole process

Table S10. 25(OH)D deficiency in PD versus control

Authors and year

Country

Study Type

Sample size

Age (mean)

Intervention/Treatment

Engagement/Adherence (%)

Study Duration

(months or weeks)

25(OH)D serum concentration changes at baseline and endpoints [units; mean (SD)]

OR (95% CI)

Control Group

Intervention group

Control Group

Intervention group

Control Group

Intervention group

Control Group

Intervention group

XXX et al., year [reference]

Number of participants

Number of participants

Mean age

Mean age

% of participants that remained the whole process

Table S11. VitD and PD incidence

Authors and year

Country

Study Type

Sample size

Age (mean)

Intervention/Treatment

Engagement/Adherence (%)

Study Duration

(months or weeks)

25(OH)D serum concentration changes at baseline and endpoints [units; mean (SD)]

HR (95% CI)

Control Group

Intervention group

Control Group

Intervention group

Control Group

Intervention group

Control Group

Intervention group

XXX et al., year [reference]

Number of participants

Number of participants

Mean age

Mean age

% of participants that remained the whole process

Table S12. VitD supplementation versus placebo in PD.

Authors and year

Country

Study Type

Sample size

Age (mean)

Intervention/Treatment

Engagement/Adherence (%)

Study Duration

(months or weeks)

UPDRS- Total “ON”(Δ) UPDRS-III during “ON” (Δ) TUG “ON”(Δ)

UPDRS- Total “ON”(Δ) UPDRS-III during “ON” (Δ) TUG “ON”(Δ)

HR (95% CI) UPDRS- Total “ON”(Δ) UPDRS-III during “ON” (Δ) TUG “ON”(Δ)

Control Group

Intervention group

Control Group

Intervention group

Control Group

Intervention group

XXX et al., year [reference]

Number of participants

Number of participants

Mean age

Mean age

% of participants that remained the whole process

Table S14. Effects of VitD in In Vitro Models of PD and Table S15. Effects of VitD in In Vivo Models of PD :

Authors and year

Country

Study Type

Sample size

Age (mean)

Intervention/Treatment

Engagement/Adherence (%)

PD Model

Study Duration

(months or weeks)

Observed Outcome

Control Group

Intervention group

Control Group

Intervention group

Control Group

Intervention group

XXX et al., year [reference]

Number of participants

Number of participants

Mean age

Mean age

VitD form & dose

VitD form & dose

% of participants that remained the whole process

Table S17. Clinical trials assessing the effect of VitD on PD.

Authors and year

Country

Study Type

Sample size

Age (mean)

Intervention/Treatment

Start

Primary outcome Secondary outcome

Primary outcome Secondary outcome

Control Group

Intervention group

Control Group

Intervention group

Control Group

Intervention group

XXX et al., year [reference]

Number of participants

Number of participants

Mean age

Mean age

VitD form & dose

VitD form & dose

Discussion:

Line 369: "Zhang et al. " Please, add the reference in authors citation throughout the text when you have followed the previous example using Zhang et al., [reference XXX].

The references were cited, at the last presentation of the information to avoid confusion, but we can modify if the reviewer believes that this is better interpreted.

Figure 8 could benefit from higher resolution.

We used the maximum resolution available at Power Point. Also, the information can be found in tables S14 and S15.

Line 513: In vitro should be in italics.

Line 526: "In both cellular and C. elegans PD models, (...)" When mentioning the first time C. elegans should be by extense not in abbreviature. Also, it should be in italics.

Line 535: "are a promising in vitro ". In vitro should be in italics.

Line 536: "Compared to in vivo". In vivo should be in italics.

Line 546: In Vivo should be in italics. Please verify throughout the manuscript and suppelementar material and correct it.

Nomenclature, and italics were added to address this comment.

Line 721: 4.7. Amount of VITD and Recommendations

The current recommendations for general population is very well described and synthetized in Table 3, however I kindly recommend the authors to introduce more information about if the supplementation with Vit D in patients with PD has already been advised by official sources (from UK for example, since the authors mention United Kingdom Parkinson’s Disease Brain Bank) and if so, which dose prescription has been recommended, at which stages of the diagnosis and which improvements have been proved associated to this intake.

Thank you for your suggestion. We reviewed official UK guidelines and found that while general recommendations for vitamin D supplementation exist (e.g., 10 µg/day for adults as per Public Health England), there are currently no specific national guidelines advising vitamin D supplementation exclusively for Parkinson’s disease patients. Clinical practice generally follows standard population guidance unless deficiency is confirmed. Some studies have explored supplementation in PD, typically using doses ranging from 800 IU to 1,200 IU/day, reporting improvements in bone health and, in some cases, motor function scores, but these findings are not yet incorporated into formal UK or international PD-specific guidelines. We will clarify this in the manuscript and cite relevant sources.

Overall, the work is well designed and brings novelty to the state of art making a stand of the current situation on the matter, syntethizing the global information gathered through diversified line of studies, impartially outweighting the strengths and limiations found in them.

Round 2

Reviewer 1 Report

Comments and Suggestions for Authors

Dear Authors, 

We acknowledge and appreciate the authors' work in responding to reviewers' recommendations and addressing queries, which has greatly enhanced the paper. Your work is well-crafted and has contributed new perspectives to the literature. You have successfully improved the manuscript, making it suitable for publication in NeuroSci.

Best Regards

Author Response

Dear Reviewer,
Thank you very much for your kind and encouraging feedback. We are truly grateful for your thoughtful review and are pleased to hear that our revisions have enhanced the manuscript. Your recognition of our work and its contribution to the literature is deeply appreciated.
Sincerely,
Authors

Reviewer 2 Report

Comments and Suggestions for Authors

Concerns were addressed defensively without substantive revision; key methodological issues remain unresolved.

Author Response

Dear Reviewer,
Thank you for your continued engagement and for highlighting the remaining concerns regarding our manuscript. We sincerely apologize if our previous responses appeared defensive rather than constructive. We greatly value your insights and recognize the importance of addressing methodological issues thoroughly to strengthen the rigor and clarity of our work.

Sincerely,

Authors

Reviewer 4 Report

Comments and Suggestions for Authors

Dear Authors,

congratulations on the improvement of your manuscript.

I would just recommend reviewing the citations of colleagues. Eg.:

Derex et al., please cite as Derex et al., [number of reference].

Please, verify that for all the 118 citations throughout the manuscript.

Author Response

We appreciate the reviewer's comment, and the references were checked.